# Neuron-specific inactivation of *Wt1* alters locomotion in mice and changes interneuron composition in the spinal cord

Danny Schnerwitzki[1], Sharn Perry[2], Anna Ivanova[1], Fabio V Caixeta[2], Paul Cramer[1], Sven Günther[1], Kathrin Weber[1], Atieh Tafreshiha[2], Lore Becker[3], Ingrid L Vargas Panesso[3,6], Thomas Klopstock[6,7,8,9], Martin Hrabe de Angelis[3,10,11], Manuela Schmidt[4], Klas Kullander[2], Christoph Englert[1,5]

Locomotion is coordinated by neuronal circuits of the spinal cord. Recently, dI6 neurons were shown to participate in the control of locomotion. A subpopulation of dI6 neurons expresses the Wilms tumor suppressor gene *Wt1*. However, the function of Wt1 in these cells is not understood. Here, we aimed to identify behavioral changes and cellular alterations in the spinal cord associated with *Wt1* deletion. Locomotion analyses of mice with neuron-specific *Wt1* deletion revealed a slower walk with a decreased stride frequency and an increased stride length. These mice showed changes in their fore-/hindlimb coordination, which were accompanied by a loss of contralateral projections in the spinal cord. Neonates with *Wt1* deletion displayed an increase in uncoordinated hindlimb movements and their motor neuron output was arrhythmic with a decreased frequency. The population size of dI6, V0, and V2a neurons in the developing spinal cord of conditional *Wt1* mutants was significantly altered. These results show that the development of particular dI6 neurons depends on *Wt1* expression and that loss of *Wt1* is associated with alterations in locomotion.

## Introduction

In vertebrates, rhythmic activity is generated by a network of neurons, commonly referred to as central pattern generators (CPGs) (Jessell, 2000; Grillner, 2003; Kiehn, 2006; Brownstone & Wilson, 2008; Goulding, 2009; Berkowitz et al, 2010). CPGs do not require sensory input to produce rhythmic output; however, the latter is crucial for the refinement of CPG activity in response to external cues (Rossignol & Drew, 1988; Jessell, 2000; Pearson, 2004). The locomotor CPGs are located in the spinal cord and consist of distributed networks of interneurons and motor neurons (MNs), which generate an organized motor pattern during repetitive locomotor tasks such as walking and swimming (Grillner, 1985; Kiehn 2006, 2016; Brownstone & Wilson, 2008; McCrea & Rybak, 2008; Goulding, 2009; Grillner & Jessell, 2009).

The spinal cord develops from the caudal region of the neural tube. The interaction of secreted molecules, including sonic hedgehog and bone morphogenetic proteins, provides instructive positional signals to the 12 progenitor cell domains that reside in the neuroepithelium (Alaynick et al, 2011). Each domain is characterized by the expression of specific transcription factor–encoding genes that are used to selectively identify these populations. The dI1–dI5 interneurons are derived from dorsal progenitors and primarily contribute to sensory spinal pathways. The dI6, V0–V3 interneurons, and MN arise from intermediate or ventral progenitors and are involved in the locomotor circuitry (Goulding, 2009).

The involvement of V0–V3 neurons in locomotion has been well documented: V0 (Lanuza et al, 2004; Talpalar et al, 2013; Bellardita & Kiehn, 2015), V1 (Zhang et al, 2014; Britz et al, 2015), V2a (Crone et al 2008, 2009; Dougherty & Kiehn, 2010; Zhong et al, 2010), and V3 (Zhang et al, 2008). The role for dI6 neurons in locomotion has only recently been addressed (Andersson et al, 2012; Dyck et al, 2012; Haque et al, 2018). A fraction of the dI6 population consists of rhythmically active neurons (Dyck et al, 2012), and a more defined subpopulation of dI6 neurons expressing the transcription factor

[1]Molecular Genetics Lab, Leibniz Institute on Aging—Fritz Lipmann Institute, Jena, Germany    [2]Department of Neuroscience, Uppsala University, Uppsala, Sweden    [3]German Mouse Clinic, Institute of Experimental Genetics, Helmholtz Zentrum München, German Research Center for Environmental Health, Neuherberg, Germany    [4]Institute of Systematic Zoology and Evolutionary Biology with Phyletic Museum, Friedrich Schiller University Jena, Jena, Germany    [5]Institute of Biochemistry and Biophysics, Friedrich-Schiller-University Jena, Jena, Germany    [6]Department of Neurology, Friedrich-Baur-Institut, Ludwig Maximilian University Munich, Munich, Germany    [7]German Center for Neurodegenerative Diseases, Munich, Germany    [8]Munich Cluster for Systems Neurology, Adolf-Butenandt-Institut, Ludwig Maximilian University Munich, Munich, Germany    [9]German Center for Vertigo and Balance Disorders, University Hospital Munich, Campus Grosshadern, Munich, Germany    [10]Chair of Experimental Genetics, School of Life Science Weihenstephan, Technical University of Munich, Freising, Germany    [11]German Center for Diabetes Research, Neuherberg, Germany

Correspondence: Christoph.englert@leibniz-fli.de

*Dmrt3* is critical for normal development of coordinated locomotion (Andersson et al, 2012). A group of dI6 neurons is suggested to express the Wilms tumor suppressor gene *Wt1* (Goulding, 2009; Andersson et al, 2012).

*Wt1* encodes a zinc finger transcription factor that is inactivated in a subset of Wilms tumors, a pediatric kidney cancer (Call et al, 1990; Gessler et al, 1990). Wt1 fulfills a critical role in kidney development; however, the function of Wt1 is not limited to this organ. Phenotypic anomalies of *Wt1* knockout mice can be found, among others, in the gonads, heart, spleen, retina, and olfactory system (Kreidberg et al, 1993; Herzer et al, 1999; Moore et al, 1999; Wagner et al 2002, 2005). In one of the first reports on *Wt1* expression, a particular region of the hindbrain below the fourth ventricle and the spinal cord were described as prominent Wt1+ tissues (Armstrong et al, 1993; Rackley et al, 1993). Very recent work focusing on *Wt1*-expressing cells in the spinal cord suggested those cells to be involved in locomotion (Haque et al, 2018). However, until now, there is no insight on the way that Wt1 determines the character of these cells.

Here, we have examined the importance of Wt1 for the developing spinal cord neurons. We performed locomotor analyses of conditional *Wt1* knockout mice and used molecular biological and electrophysiological approaches to elucidate the role that Wt1 exerts on spinal cord neurons for locomotion. Our data suggest that *Wt1*-expressing dI6 neurons contribute to the coordination of locomotion and that Wt1 is needed for proper dI6 neuron specification during development.

## Results

### *Wt1*-expressing cells in the spinal cord are dI6 neurons

To determine the spatial and temporal pattern of *Wt1*-expressing cells in the spinal cord, we performed immunohistochemical analyses. Wt1+ cells were detected in the medioventral mantle zone of the developing spinal cord at embryonic day (E) 12.5 (Fig 1A). Until E15.5, embryonic spinal cords showed a constant amount of Wt1+ cells; thereafter, their number gradually decreased until they could no longer be detected in adult mice (Fig 1B).

We next wanted to determine the birthdate of Wt1+ cells, defined as the time point when progenitor cells cease to proliferate, leave

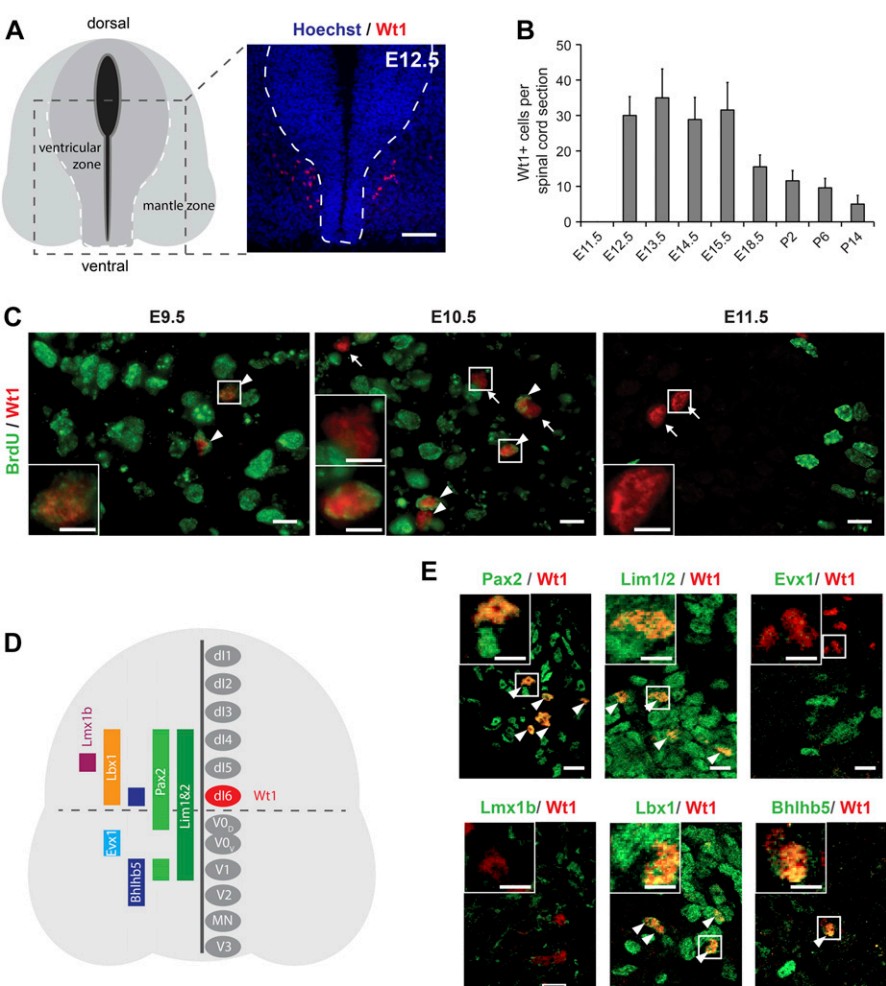

**Figure 1. Characterization of Wt1+ neurons in the developing spinal cord.**
**(A)** Schematic illustration and Wt1 immunolabelling analysis of a transverse section (12 $\mu$m) from E12.5 spinal cord showing the position of Wt1+ neurons (red) in the mantle zone of the developing spinal cord. Stippled line represents the border between the ventricular and mantle zones. Scale bar: 50 $\mu$m. **(B)** Plot showing the average cell number of Wt1+ neurons per 12 $\mu$m spinal cord section from different embryonic and postnatal stages. Wt1+ neurons are first found at E12.5 and decrease in cell number postnatally. Data expressed as mean ± SD. n = 12–20 embryos. **(C)** Determination of the birthdate of Wt1+ neurons by BrdU proliferation assay. Proliferating cells situated in the ventricular zone were labelled by BrdU incorporation at different embryonic stages (E9.5, E10.5, and E11.5). Additional immunolabelling of these cells for Wt1 and BrdU at E12.5 revealed that prospective *Wt1*-expressing cells still proliferate at E9.5 and at E10.5 but not at E11.5. Scale bar: 10 $\mu$m. Insets show higher magnifications of respective areas. Scale bar: 5 $\mu$m. **(D)** Schematic illustration of an E12.5 spinal cord section with markers and their occurrence in different neuron populations. These markers were used to establish the origin of Wt1+ neurons as dI6 neurons (red). **(E)** Immunolabelling of Wt1+ neurons with markers present in dI6 and adjacent interneuron populations. The partly overlapping location of Wt1 with Pax2, Lim1/2, Lbx1, and Bhlhb5 supports a dI6 character. Scale bar: 10 $\mu$m. Insets show higher magnifications of respective areas. Scale bar: 5 $\mu$m.

the ventricular zone, and start to differentiate. Using BrdU, the proliferative cells in the ventricular zone were labelled at different embryonic stages (E9.5, E10.5, and E11.5). Immunostaining of these cells for Wt1 at E12.5 revealed that prospective *Wt1*-expressing cells still proliferate at E9.5 and even at E10.5 (Fig 1C). At E11.5, Wt1+ cells no longer showed incorporation of BrdU, suggesting that they had left the ventricular zone and started their migration and differentiation in the mantle zone at this time point.

Wt1 has been proposed to label dI6 neurons (Goulding, 2009); however, the only available primary data have so far only suggested its presence in a subpopulation of dI6 neurons expressing *Dmrt3* (Andersson et al, 2012). To closely examine the nature of Wt1+ cells, we performed immunostainings of embryonic spinal cords at E12.5. All cells expressing *Wt1* were positive for Pax2 and Lim1/2 labelling dI4, dI5, dI6, V0$_D$, and V1 neurons (Tanabe & Jessell, 1996; Burrill et al,

1997), while being negative for the postmitotic V0$_V$ marker Evx1 (Moran-Rivard et al, 2001) (Fig 1D and E). *Wt1* expression did not overlap with Lmx1b, a marker specific for dI5 neurons, but all Wt1+ cells exhibited Lbx1 (Gross et al, 2002) and Bhlhb5 labelling (Skaggs et al, 2011), which commonly occur in the ventral most dI4–dI6 Lbx1+ domain giving rise to dI6 neurons. Thus, these data support and extend on the previous observations that Wt1 is a marker for a subset of dI6 neurons.

### Deletion of *Wt1* affects locomotor behavior

Because a constitutive knockout of *Wt1* is embryonically lethal, we made use of a conditional *Nes-Cre;Wt1$^{fl/fl}$* mouse line to investigate the function of Wt1 in the spinal cord (Fig 2A). At E12.5, no *Wt1* mRNA or protein was detected in neurons from this mouse line (Fig 2A and B).

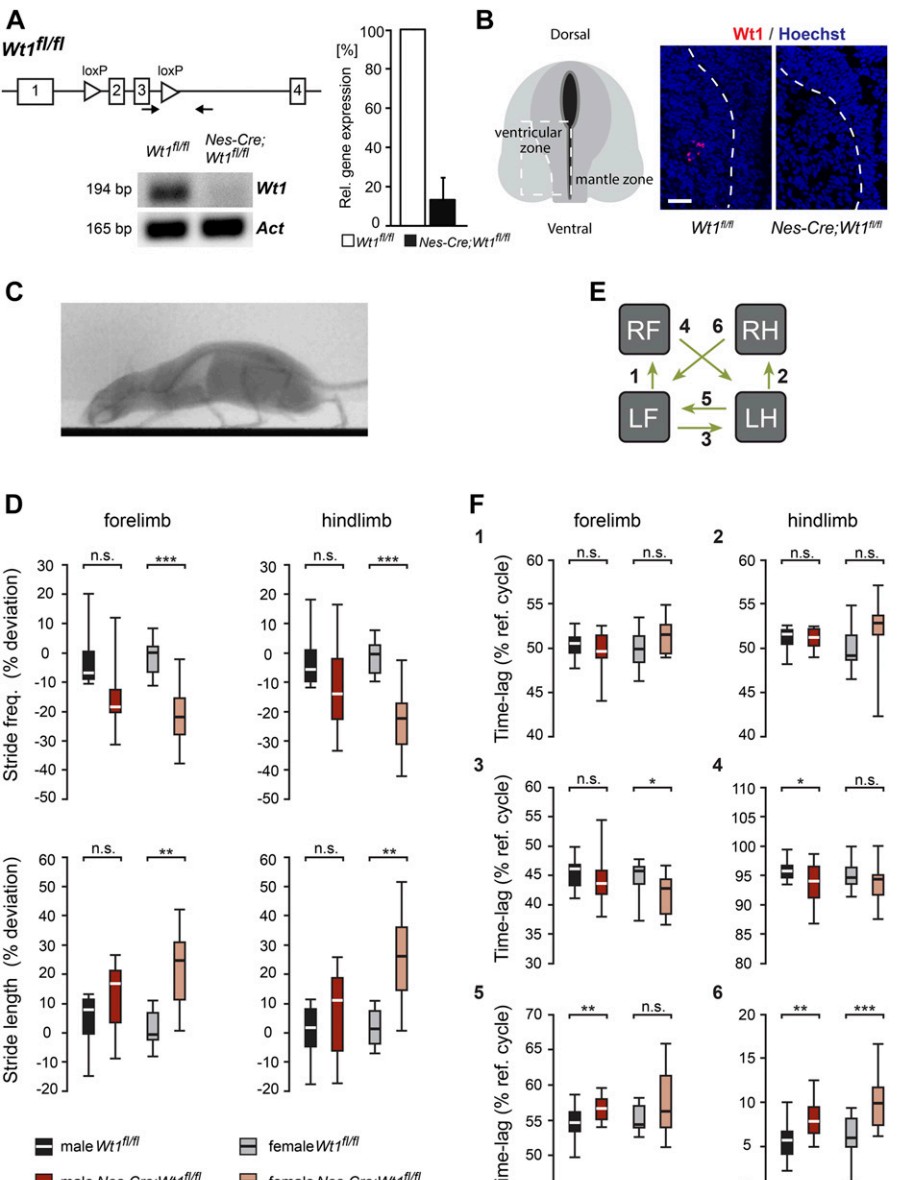

**Figure 2.  Mice with *Wt1* inactivation display altered locomotion.**
**(A)** Schematic illustration of the *Wt1$^{fl/fl}$* allele. *loxP* sites flanking exons 2 and 3 of the *Wt1* coding sequence allow Cre-mediated excision and conditional knockout of *Wt1*. Confirmation of a functional conditional *Wt1* knockout in *Nes-Cre;Wt1$^{fl/fl}$* at E12.5 using qRT–PCR (quantification to the right). Data expressed as mean ± SEM. n = 4–5 embryos. Significance determined by using pairwise reallocation randomisation test. **(B)** Loss of Wt1 immunopositive signals in *Nes-Cre;Wt1$^{fl/fl}$* embryos at E12.5 corroborates the loss of Wt1 protein. Schematic illustration shows the position where the pictures were taken. Stippled line represents the border between the ventricular and mantle zones. Scale bar: 40 $\mu$m. **(C)** X-ray radiograph of a walking mouse in lateral perspective. **(D)** Graphs displaying stride parameters collected in the X-ray radiograph. Stride frequency is significantly lower in female *Nes-Cre;Wt1$^{fl/fl}$* mice in both forelimbs and hindlimbs. The stride length in female *Nes-Cre;Wt1$^{fl/fl}$* mice is increased compared with female *Wt1$^{fl/fl}$* mice, whereas smaller differences are found in male mice. Box plots indicate the median of each group, n = 10 animals (bold white or black line), the 25th and the 75th percentile (box), and the data range (whiskers). Mann–Whitney *U* test was performed. Significance level of *U*: ***$P < 0.001$; **$P < 0.01$; *$P < 0.05$. **(E, F)** Interlimb coordination expressed as the time lag between footfall events in percent stride duration. Left limbs are reference limbs. The scheme (E) illustrates which phase relationships are shown by which graph. Phase relationships (F) between forelimbs (1) and between hindlimbs (2) illustrate overall symmetry of the walk. Timing of forelimb touchdown relative to hindlimb touchdown for ipsilateral (3) and contralateral (4) limbs show only minor differences between *Wt1$^{fl/fl}$* and *Nes-Cre;Wt1$^{fl/fl}$* mice. The timing of hindlimb footfalls relative to forelimb footfalls (5, 6) differ between *Wt1$^{fl/fl}$* and *Nes-Cre;Wt1$^{fl/fl}$* mice, particularly at the contralateral limbs. Box plots indicate the median of each group, n = 10 animals (bold white or black line), the 25th and the 75th percentile (box), and the data range (whiskers). Mann–Whitney *U* test or *t* test. Significance level of *U* or t$_s$: ***$P < 0.001$, **$P < 0.01$, *$P < 0.05$.

Given the location of the Wt1+ neurons within the ventral dI6 population that has been shown to be involved in regulating locomotion, we performed behavioral tests associated with loco-motion to investigate potential phenotypic consequences of de-leting *Wt1* in spinal cord neurons. Footprints of adult mice walking on a transparent treadmill at fixed speeds (0.15, 0.25, and 0.35 m/s) were recorded to analyze different gait parameters (Fig S1A). *Nes-Cre;Wt1$^{fl/fl}$* mice revealed a significant reduction in stride frequency for both the fore- and hindlimbs relative to control (*Wt1$^{fl/fl}$*) animals at all speeds measured. Heterozygous *Wt1* knockout mice (*Nes-Cre; Wt1$^{fl/+}$*) did not differ significantly from controls. Stride length, accordingly, was significantly longer in *Nes-Cre;Wt1$^{fl/fl}$* animals than in wild-type mice and *Nes-Cre;Wt1$^{fl/+}$*. Thus, although *Nes-Cre;Wt1$^{fl/fl}$* mice were slightly smaller than controls (body mass *Wt1$^{fl/fl}$* versus *Nes-Cre;Wt1$^{fl/fl}$*: males, 33 ± 3.9 versus 25 ± 3.7 g; females, 25 ± 3.2 g versus 22 ± 1.4 g; body length: males, 9.9 ± 0.4 g versus 9.4 ± 0.4 cm; females, 9.9 ± 0.4 cm versus 9.8 ± 0.3 cm), they made longer strides with lower frequency.

To further explore gait alterations, we used X-ray fluoroscopy as a complementary method in a larger cohort of mice (Figs 2C and S1B and Videos 1 and 2). When animals walked voluntarily at their preferred speed, deviations in stride frequency and stride length from the expected value (control baseline) for the given speed were again observed in *Nes-Cre;Wt1$^{fl/fl}$* (Fig 2D), but statistical signifi-cance is confirmed only for females. The changes were accom-panied by a significant reduction of raw speed (animal velocity in m/s) and size-corrected speed (= Froude number) in *Nes-Cre;Wt1$^{fl/fl}$* mice of both sexes (Fig S1C). Although both the duration of stance and swing phases and the distance covered by the trunk and the limbs, respectively, differ between controls and *Nes-Cre;Wt1$^{fl/fl}$* by more than 10 percent in males and more than 15 percent in females, the ratio between the two phases, expressed by the duty factor, re-mains unaffected (Fig S1D). Thus, the temporal coordination between stance and swing phases in adult *Nes-Cre;Wt1$^{fl/fl}$* mice is normal.

We tested whether changes in gait parameters are accompanied by changes in the phase relationships between the limbs (Fig 2E and F). The footfall pattern of control and *Nes-Cre;Wt1$^{fl/fl}$* females did not show significant differences at the same speed of 0.21 m/s (Fig S1E). However, the different spread along the X-axis indicates the evenly elongated stance and swing phases.

The symmetry of left and right limb movements expressed as the time lag between footfalls in percent stride duration of a reference limb (Fig S1F) was unaffected in the *Nes-Cre;Wt1$^{fl/fl}$* mice (Fig 2E and F, 1 and 2). Also, the timing of forelimb footfalls relative to the ipsilateral and contralateral hindlimb cycles is very similar be-tween *Wt1$^{fl/fl}$* mice and *Nes-Cre;Wt1$^{fl/fl}$* mice (Fig 2E and F, 3 and 4). Significant differences between *Wt1$^{fl/fl}$* mice and *Nes-Cre;Wt1$^{fl/fl}$* mice were observed in the timing of the hindlimb footfalls relative to the forelimb cycles (Fig 2E and F, 3 and 4). The touchdown of the ipsilateral and the contralateral hindlimb falls in a later fraction of the forelimb stride cycle in *Nes-Cre;Wt1$^{fl/fl}$* mice compared with the *Wt1$^{fl/fl}$* mice. The deviation cannot be explained by the differences in animal speed because the hind-to-forelimb coordination shows only small amount of speed-dependent variation: the time lag between the footfalls tend to increase with increasing speed (baseline ipsilateral: *Wt1$^{fl/fl}$* males: F$_{1,248}$ = 13.38, r$^2$ = 0.051, *Wt1$^{fl/fl}$* females: F$_{1,248}$ = 18.63, r$^2$ = 0.070; baseline contralateral: *Wt1$^{fl/fl}$*

males: F$_{1,273}$ = 16.39, r$^2$ = 0.057, *Wt1$^{fl/fl}$* females: F$_{1,274}$ = 8.14, r$^2$ = 0.029). So far, the limb kinematics of adult *Nes-Cre;Wt1$^{fl/fl}$* mice compared with the *Wt1$^{fl/fl}$* mice shows subtle differences in gait parameters and interlimb coordination with a high degree of variation. In sum, these differences result in a performance reduction indicated by the overall lower walking velocities.

## Deletion of *Wt1* results in a disturbed and irregular postnatal locomotor pattern

After having observed altered gait parameters in adult *Nes-Cre; Wt1$^{fl/fl}$* animals, we wondered whether gait also would be affected in younger mice. Indeed, *Nes-Cre;Wt1$^{fl/fl}$* pups had more difficulty coordinating their fore- and hindlimbs than controls when per-forming air-stepping. Although there was no increase in hind-limb synchronous steps, left/right alternating steps were decreased and the number of uncoordinated steps was increased in *Nes-Cre;Wt1$^{fl/fl}$* animals (Fig S2 and Videos 3 and 4). We next performed fictive locomotion experiments on isolated spinal cords from control and *Nes-Cre;Wt1$^{fl/fl}$* mice (P0–P3). Fictive locomotor drugs induced a markedly slower, disturbed, more variable pattern of locomotor-like activity in *Nes-Cre;Wt1$^{fl/fl}$* spinal cords than the stable, rhythmic pattern of locomotor-like activity in control mice. Control spinal cords had recorded activity bursts that showed clear left/right (L2 versus L2) and flexor/extensor (L2 versus L5) alternation that persisted throughout activity periods, whereas activity bursts in *Nes-Cre;Wt1$^{fl/fl}$* spinal cords were uncoordinated and did not maintain strict left/right or flexor/extensor alternation (Fig 3A and B). The relationship between left/right and flexor/extensor alter-nation was examined, and control cords presented a reliable phase preference around 180° (Fig 3C; control average phase pref-erence in l/r: 183.4°, R = 0.93; in f/e: 185.2°, R = 0.84). However, spinal cords from mice with *Wt1* deletion showed an irregular locomotor pattern with inconsistent alternation as indicated by its short-phase vector (Fig 3C; *Wt1$^{fl/fl}$* average phase preference in l/r: 165.3°, R = 0.60; in f/e: 155.2°, R = 0.44). Although there was no difference in the preferred phase across the two groups (l/r Watson's U$^2$ = 0.10, P > 0.05; f/e Watson's U$^2$ = 0.07, P > 0.05), the coupling strength, or R, as indicated by the vector length in the polar plots, was significantly decreased upon *Wt1* deletion (l/r P = 0.031 and f/e P = 0.002, one-tailed Mann–Whitney *U* test). In addition, the frequency of the ventral root output was decreased (Fig 3D: control; 0.30 ± 0.024 Hz: *Nes-Cre; Wt1$^{fl/fl}$*; 0.18 ± 0.08 Hz). This slower rhythm in *Nes-Cre;Wt1$^{fl/fl}$* cords could be attributed to altered L2 and L5 activity burst parameters, as *Nes-Cre;Wt1$^{fl/fl}$* mice had significantly longer burst, interburst, and cycle periods than control (Fig 3E and F). Thus, the deletion of *Wt1* results in a disturbed and irregular locomotor pattern, which suggests that there are changes to the neuronal locomotor circuitry that occur following *Wt1* deletion.

## Wt1+ neurons receive various synaptic inputs and can project commissurally

To assess how Wt1+ dI6 neurons are connected within the CPG network, we focused on the innervation pattern of these cells. We used the *Wt1-GFP* reporter mouse line (Hosen et al, 2007) where Wt1+ neurons are labelled by GFP. In contrast to the restricted

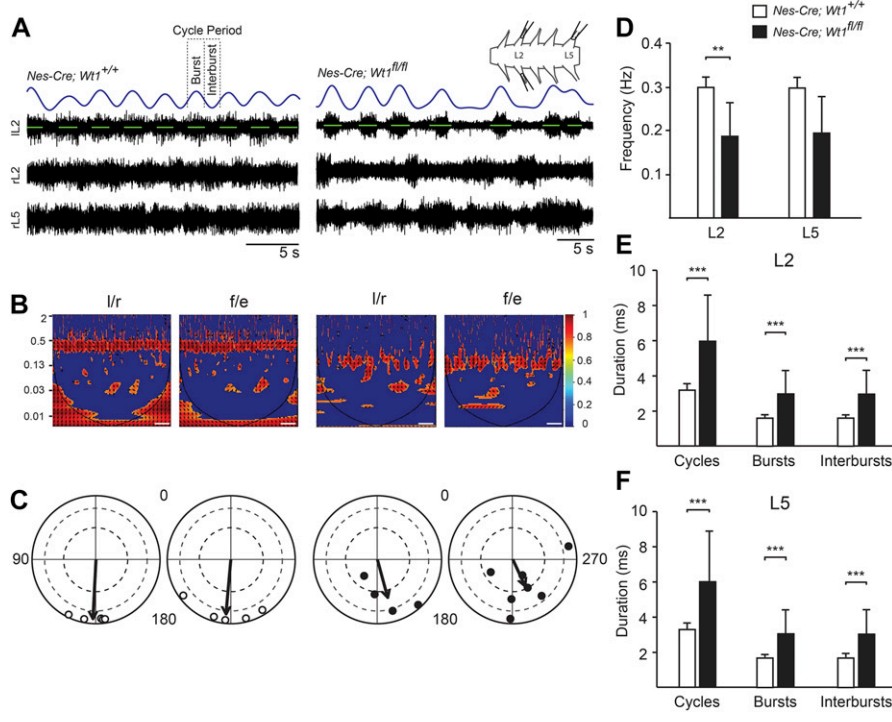

**Figure 3. Locomotor activity is variable and uncoordinated in *Nes-Cre; Wt1^{fl/fl}* pups.**
**(A)** Representative traces showing locomotor-like activity during fictive locomotion from left and right lumbar (L) 2 and right L5 ventral roots from control (*Nes-Cre;Wt1^{+/+}*) and *Wt1* conditional knockout (*Nes-Cre; Wt1^{fl/fl}*) mice. Rhythmic activity was induced by application of NMDA, serotonin, and dopamine. Raw traces in black; rectified, low-pass filtered signal of lL2 trace in blue; activity burst shown in green. Spinal cord schematic depicts the attached suction electrodes to the right (r) and left (l) L2 and rL5 ventral roots. Scale bar: 5 s. **(B)** Phase analysis and associated coherence power spectra of left/right (L2/L2) and flexor/extensor (L2/L5) recordings. Regions of persistent coherence emerge for control mice at 0.30 Hz, whereas spinal cords from *Nes-Cre;Wt1^{fl/fl}* mice show an intermittent coherence region at 0.18 Hz. Color-graded scale indicates normalized coherence. Scale bar: 125 s. **(C)** Locomotor patterns, analyzed from 20 consecutive bursts, reveal impaired and variable left/right and flexor/extensor alternation in *Nes-Cre;Wt1^{fl/fl}* mice (black dots). Normal left/right and flexor/extensor alternation is maintained in control (white dots) mice. Each dot represents one cord; arrows represent the mean phase. The length of the vector is a measure of the statistical significance of the preferred phase; dashed grey line indicates regions of significance and high significance at 0.5 and 0.8, respectively (Rayleigh test and Watson's $U^2$ test). n = 5 pups (control); n = 7 pups (*Nes-Cre;Wt1^{fl/fl}*). **(D)** *Nes-Cre;Wt1^{fl/fl}* mice have a slower locomotor frequency than control mice. Data are shown as mean ± SD. n = 5 pups (control); n = 7 pups (*Nes-Cre;Wt1^{fl/fl}*). Significance was tested using two-tailed Mann–Whitney *U* test. **(E, F)** The slower locomotor frequency in *Nes-Cre;Wt1^{fl/fl}* mice is mirrored by an increased cycle period, and burst and interburst duration in both L2 (E) and L5 (F) roots. Data are shown as mean ± SD. n = 5 pups (control); n = 7 pups (*Nes-Cre;Wt1^{fl/fl}*). Significance was tested using two-tailed Mann–Whitney *U* test. Significance level: **P < 0.01, ***P < 0.001.

localization of Wt1 in the nucleus, GFP is distributed throughout the cytoplasm and labels the soma and major processes (Fig 4A). In combination with antibodies against particular vesicular synaptic transporters, we observed that excitatory (VGLUT2), inhibitory (VGAT), and modulatory (VMaT2) synapses contact the soma of Wt1+ dI6 neurons (Fig 4B). This shows that Wt1+ dI6 neurons receive excitatory, inhibitory, and modulatory inputs, suggesting that Wt1+ neurons are positioned to receive a multitude of signals and could act during locomotion to integrate different CPG signals.

Using the *Wt1-GFP* reporter mouse, we found GFP+ fibers crossing the spinal cord midline beneath the central canal, suggesting that Wt1+ neurons project commissural fibers (Fig 4C). Fluorescent dextran amine retrograde tracing of contralateral projections confirmed that at least part of the Wt1+ dI6 neurons project commissurally (Fig S3). We analyzed spinal cord commissural neurons in control (Fig 4D, left) and homozygous *Wt1*-mutant (Fig 4D, right) mice (P1–5) to determine whether the deletion of *Wt1* alters the total number of commissural neurons and investigated ascending (aCIN), descending (dCIN), and bifurcating (adCIN) subpopulations (Fig 4D and E). All traced subpopulations were markedly reduced in *Nes-Cre;Wt1^{fl/fl}* spinal cords compared with controls (Fig 4F–H), which suggests that Wt1 is crucial for proper axonal projection pattern.

## Loss of Wt1 leads to altered interneuron composition

To assess the possible impact of *Wt1* deletion for interneuron development, we analyzed dI6 and non-dI6 populations situated in the embryonic ventral spinal cord. The number of *Dmrt3*-expressing cells, which constitutes a distinct but partly overlapping dI6 population (Andersson et al, 2012), was significantly decreased in the embryos harboring a loss of Wt1 in the spinal cord already at E12.5 (Fig 5A) persisting throughout development (E16.5 and P1). At any investigated time point, neurons co-expressing both *Wt1* and *Dmrt3* were not detected in *Nes-Cre;Wt1^{fl/fl}* embryos and neonates.

Loss of the transcription factor Dbx1 that is involved in differentiation of the V0 population results in a fate switch of some V0 neurons to become dI6 interneuron-like cells (Lanuza et al, 2004). Thus, we investigated whether populations flanking the dI6 population were affected in *Nes-Cre;Wt1^{fl/fl}* mice. The Lmx1b+ dI5 population was similar in number when comparing *Nes-Cre;Wt1^{fl/fl}* with wild-type embryos, whereas the number of Evx1+ V0$_V$ neurons was significantly increased already at E12.5 (Fig 5B). This increase was still detectable at E16.5. No differences could be seen in Foxp2+ V1 neurons, Chx10 (V2a) and Gata3 (V2b) neurons, and Islet 1/2+ MNs between conditional *Wt1* knockout and control embryos at E12.5. However, at E16.5, Chx10+ V2a neurons showed a significant decrease in cell number.

To verify the changes in interneuron composition found in the developing *Nes-Cre;Wt1^{fl/fl}* mice, we made use of a second mouse line, namely, *Lbx1-Cre;Wt1^{fl/fl}* mice. At embryonic stage E16.5, we observed a decrease in the amount of dI6 neurons and an increase in the cell number of Evx1+ neurons similar to *Nes-Cre;Wt1^{fl/fl}* mice (Fig 5C). This decline in the number of dI6 neurons and the concomitant increase in the amount of Evx1+ neurons might point to a change in the developmental fate from dI6 neurons into V0 neurons prompted by the deletion of *Wt1*. To test this hypothesis, we ablated the cells destined to express Wt1. We used *Lbx1-Cre;Wt1-GFP-DTA* mice in

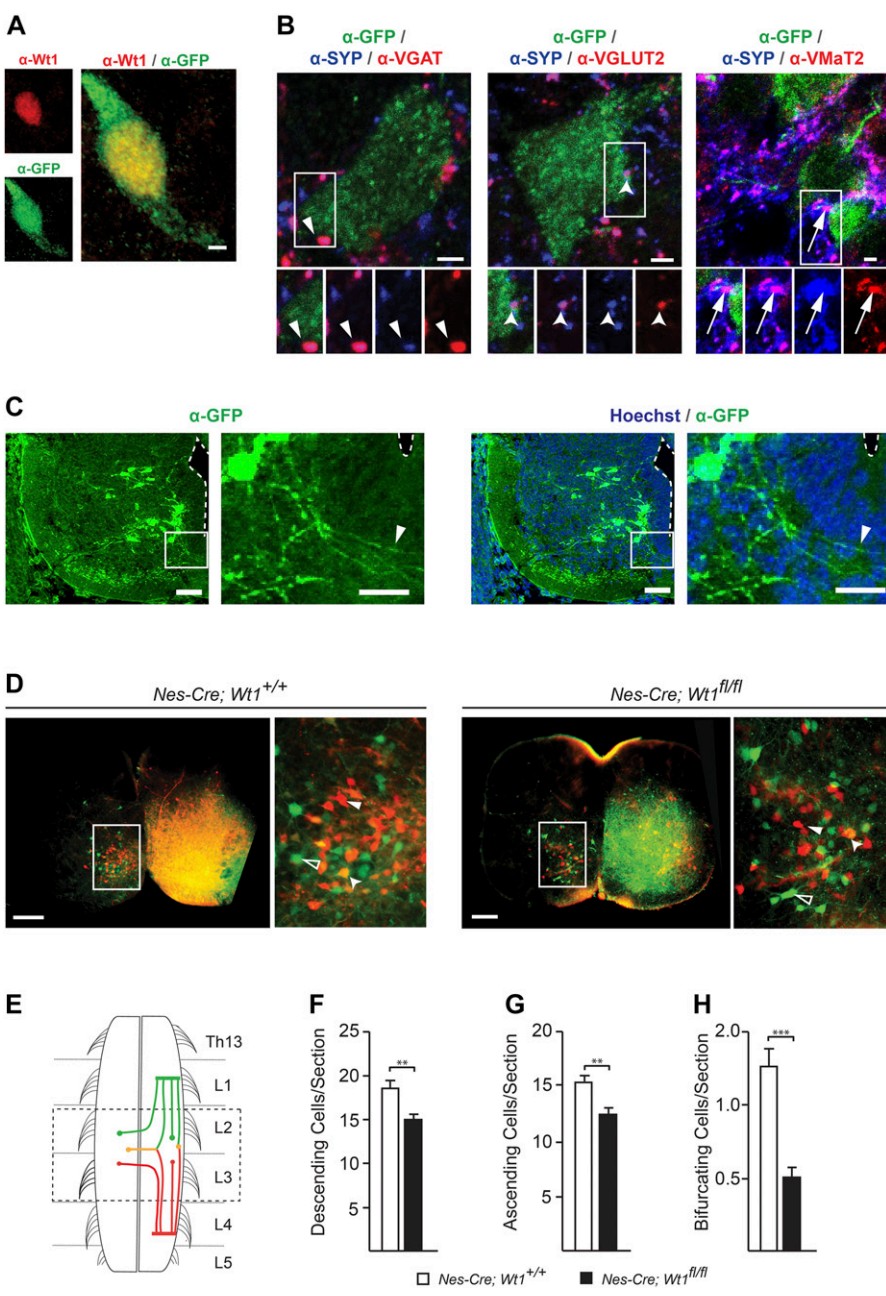

**Figure 4. Innervation of Wt1+ neurons and number of commissural neurons in neonatal mice.**
**(A)** Spinal cords of *Wt1-GFP* embryos (stage E16.5) show a Wt1+ interneuron immunopositive for GFP. Wt1 is localized in the nucleus and GFP throughout the cell. Scale bar: 2 μm. **(B)** Wt1+ neurons (green) receive excitatory, inhibitory, and monoaminergic synaptic contacts. Synaptic terminals are identified with synaptophysin (blue). Glutamatergic terminals were immunolabelled for VGLUT2, inhibitory synapses immunolabelled for VGAT, and monoaminergic terminals immunolabelled for VMAT2. Arrows point to individual synaptic terminals (magenta) present on Wt1+ neurons (green). Boxed areas show higher magnification panels of separated channels. Scale bar: 2 μm. **(C)** GFP-immunolabelled dI6 neurons in the spinal cord of E16.5 *Wt1-GFP* embryos. GFP antibody staining (green; left panel) and merging with Hoechst (blue; right panel) is shown. Boxed areas represent location of higher magnification panels shown on the right of each panel. Contralateral projections crossing the midline (dashed line) of the spinal cord are visible (arrow heads in magnified images). Scale bar: 20 μm for overview images and 50 μm for magnified images. **(D)** Photomicrographs of transverse, 60 μm, lumbar, spinal cord sections with applied fluorescein dextran amine (FDA, green) and rhodamine dextran amine (RDA, red) tracers. Higher magnification images (insets) of wild-type (*Nes-Cre;Wt1$^{+/+}$*) and homozygous (*Nes-Cre;Wt1$^{fl/fl}$*) segments, showing intersegmental retrograde FDA (white arrow), RDA (open arrow), and double-labelled (triangle arrow) neurons. Scale bar: 200 μm. **(E)** Schematic illustration of FDA (lumbar [L]1) and RDA (L4/5) application sites tracing descending (green), ascending (red), and bifurcating (yellow) neurons. The area of analysis (L2/L3) is indicated by black dashed line. **(F–H)** Quantification of descending FDA-labelled neurons (F), ascending RDA-labelled neurons (G), and bifurcating, double-labelled neurons per section (H). Descending, ascending, and bifurcating CINs are significantly fewer in homozygous spinal cords compared with control cords (according to Kruskal–Wallis test and followed by a Dunn's post hoc test comparing all groups). Data expressed as mean ± SEM (*Wt1$^{fl/fl}$* control: 3,975 total cells, 215 sections, and nine spinal cords; *Nes-Cre;Wt1$^{fl/fl}$*: 3,421 total cells, 228 sections, and seven spinal cords). Significance level: *$P < 0.05$, **$P < 0.01$, ***$P < 0.001$.

which the diphtheria toxin subunit A (*DTA*) is expressed from the endogenous *Wt1* locus after Cre-mediated excision of a *GFP* cassette harboring a translational STOP codon. *Cre* expression driven by the *Lbx1* promoter targets the dI4 to dI6 interneuron populations (Müller et al, 2002). In *Lbx1-Cre;Wt1-GFP-DTA* embryos, nearly all Wt1+ neurons were ablated at E16.5 (Fig 5D). The ablation of Wt1+ neurons coincided with a significantly decreased number of Dmrt3+ neurons in *Lbx1-Cre;Wt1-GFP-DTA* embryos, but did not affect the number of Evx1+ neurons (Fig 5D). Taken together, the results from the *Wt1* deletion and the ablation of the Wt1 neurons suggest that the fate switch from dI6 neurons into Evx1+ V0 neurons occurs because of the deletion of *Wt1*. A postnatal phenotypic behavioral analysis of these mice was not possible

because neonates died immediately after birth because of serious respiratory deficits (data not shown).

The analyses of the interneuron composition in developing conditional *Wt1* knockout mice and embryos with an ablation of Wt1+ neurons suggest a fate switch within a specific subset of dI6 and V0$_V$ neurons that depends on the presence of the cells destined to express *Wt1*.

### The transition of dI6 neurons into Evx1+ V0$_V$ neurons upon loss of *Wt1* is not direct

To further investigate the cellular fate change upon deletion of *Wt1*, we combined *Wt1-GFP* and *Nes-Cre;Wt1$^{fl/fl}$* animals to generate

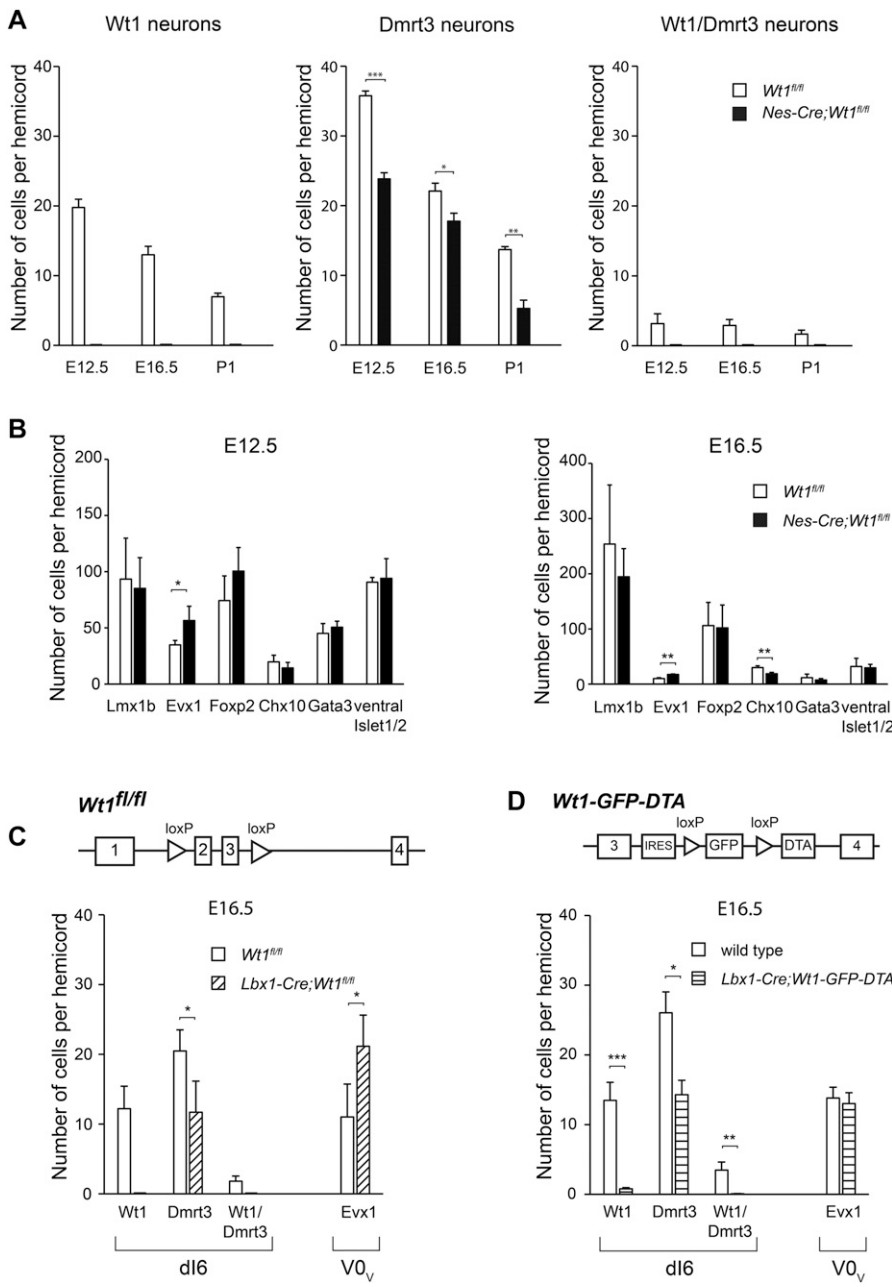

**Figure 5. Alterations in the composition of ventral neurons upon *Wt1* knockout.**
**(A)** Average cell number of Wt1, Dmrt3, and Wt1/Dmrt3 neurons per 12 μm spinal cord section from different embryonic and postnatal stages of control (*Wt1^{fl/fl}*) and *Wt1* conditional knockout (*Nes-Cre;Wt1^{fl/fl}*) mice. Number of Dmrt3 neurons significantly decrease in *Nes-Cre;Wt1^{fl/fl}*. No Wt1/Dmrt3 neurons are detected in *Nes-Cre;Wt1^{fl/fl}* animals. Data expressed as mean ± SD. n = 3 embryos per developmental stage and genotype. Significance determined by *t* test. **(B)** Average cell number of Lmx1b, Evx1, Foxp2, Chx10, Gata3, and ventral Islet1/2 neurons per 12 μm spinal cord section from control (*Wt1^{fl/fl}*) and homozygous (*Nes-Cre;Wt1^{fl/fl}*) mice at E12.5 and E16.5. Number of Evx1+ V0 neurons is significantly increased. Data expressed as mean ± SD. n = 3–4 embryos per developmental stage and genotype. Significance determined by *t* test. **(C)** Average cell number of Wt1, Dmrt3, and Wt1/Dmrt3 dI6 neurons and Evx1 V0 neurons per 12 μm spinal cord section from E16.5 control (*Wt1^{fl/fl}*) and *Wt1* conditional knockout (*Lbx1-Cre;Wt1^{fl/fl}*) mice. *Lbx1-Cre*–based conditional *Wt1* knockouts show a decrease in the amount of dI6 neurons and an increase in the cell number of Evx1 neurons as for *Nes-Cre;Wt1^{fl/fl}* animals. Data expressed as mean ± SD. n = 3–5 embryos per genotype. Significance determined by *t* test. **(D)** Schematic illustration of *Wt1-GFP-DTA* allele. Cassette consisting of *loxP* sites flanking *GFP* coding sequence upstream of *DTA* was inserted into the *Wt1* locus. This cassette allows Cre-mediated ablation of Wt1+ neurons. Graph shows average cell number of Wt1, Dmrt3, and Wt1/Dmrt3 dI6 neurons and Evx1+ V0 neurons per 12 μm spinal cord section from E16.5 wild-type control and *Lbx1-Cre;Wt1-GFP-DTA* mice. Nearly all Wt1+ neurons are absent. The number of Dmrt3 neurons is significantly decreased. Population size of Evx1 neurons is not altered after ablation of Wt1+ neurons. Data expressed as mean ± SD. n = 3 embryos per genotype. Significance determined by *t* test. Significance level: \**P* < 0.05, \*\**P* < 0.01, \*\*\**P* < 0.001.

*Nes-Cre;Wt1^{fl/GFP}* mice. These mice harbor a constitutive knockout allele of *Wt1* due to the insertion of a *GFP*-coding sequence and another conditional *Wt1* knockout allele. GFP and Wt1 were co-localized in the ventral spinal cord of *Wt1^{fl/GFP}* control animals at E13.5, whereas GFP, but not Wt1, was detected in the spinal cords of *Nes-Cre;Wt1^{fl/GFP}* embryos of the same age (Fig 6A). Thus, *Nes-Cre; Wt1^{fl/GFP}* mice allowed us to inactivate *Wt1*, whereas the cells destined to express *Wt1* are labelled by GFP.

To investigate whether *Wt1* deletion leads to apoptosis in the respective cells, TUNEL was used. TUNEL+ cells were present in the ventrolateral spinal cords of *Wt1^{fl/GFP}* control and *Nes-Cre;Wt1^{fl/GFP}* embryos (Fig 6A). However, TUNEL signals never overlapped with GFP+ dI6 neurons destined to express *Wt1*,

suggesting that *Wt1* inactivation in dI6 neurons did not result in cell death.

To find out whether cells destined to express *Wt1* would directly convert to V0_V neurons upon Wt1 inactivation, we performed immunohistochemical analyses. The presence of Dmrt3 and Evx1 in GFP+ dI6 neurons was analyzed in *Wt1^{fl/GFP}* control and *Nes-Cre; Wt1^{fl/GFP}* embryos at E12.5 (Fig 6B). The number of GFP+ cells per hemicord was determined and set to 100%. The proportion of Dmrt3+ cells was approximately 13% of all GFP+ cells in the spinal cord of E12.5 control embryos. When *Wt1* was absent, the amount of Dmrt3+ GFP cells significantly decreased to 4%. In contrast, the proportion of GFP+ dI6 neurons that also showed Evx1 staining was not changed between *Wt1^{fl/GFP}* control and *Nes-Cre;Wt1^{fl/GFP}*

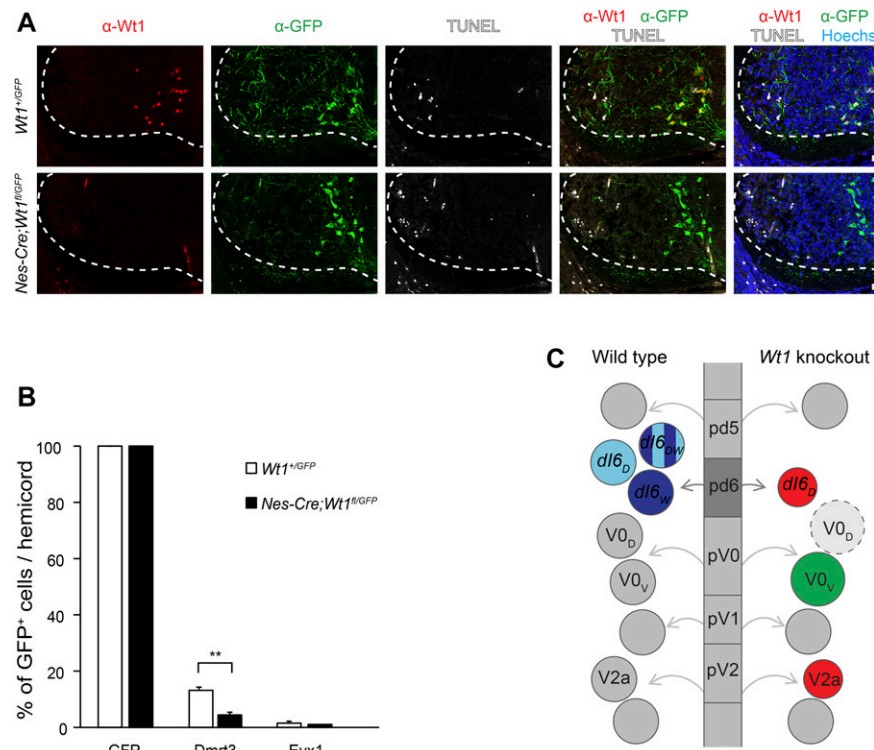

**Figure 6. Indirect trans-differentiation of Wt1+ dI6 neurons.**

**(A)** Immunofluorescence staining of spinal cord sections from E13.5 $Wt1^{+/GFP}$ (control) and $Nes$-$cre;Wt1^{fl/GFP}$ embryos. GFP is depicted in green, Wt1 in red, TUNEL+ cells in white, and Hoechst in blue. Orientation: dorsal at the top and ventral at the bottom. Scale bar: 50 $\mu m$. **(B)** Quantification of GFP+ cells harbouring the interneuron markers Dmrt3 and Evx1. Analysis was performed using E12.5 $Wt1^{+/GFP}$ (control; n = 3) and $Nes$-$cre;Wt1^{fl/GFP}$ (n = 3) embryos. The number of cells showing co-localization of GFP and the respective markers was determined and normalized to the total number of GFP+ cells, which was set to 100%. Upon $Wt1$ knockout, the amount of GFP+ cells possessing Dmrt3 is significantly decreased. The amount of GFP+ cells possessing Evx1 is not altered, suggesting an indirect fate change of dI6 neurons into Evx1+ $V0_V$-like neurons upon $Wt1$ knockout. Data expressed as mean ± SD. Significance determined by $t$ test. Significance level: $**P < 0.01$. **(C)** Scheme represents various progenitor cell domains (pd5, pd6, pV0, pV1, and pV2) that give rise to different populations of spinal cord neurons (shown as circles) in wild-type and tissue-specific $Wt1$ knockout. In wild-type animals, progenitor cells leave these domains, become postmitotic, and differentiate into distinct interneuron populations that further subdivide. The dI6 interneuron population consists of neurons either positive for Dmrt3 ($dI6_D$), Wt1 ($dI6_W$), or both ($dI6_{DW}$). Because of the knockout of $Wt1$, no $dI6_W$ and $dI6_{DW}$ are detectable and the number of $dI6_D$ neurons is decreased. In contrast, the number of Evx1+ $V0_V$ neurons increases, which is an indirect effect as potential $dI6_W$

cells that lack Wt1 did not show a Evx1 signal. This effect might be explained by a hypothetical fate change of dI6 neurons into $V0_D$-like neurons (dashed light grey circle). The increased number of $V0_D$ neurons would thus prompt the pV0 progenitor cells to differentiate preferentially into $V0_V$ neurons, which would compensate the excess amount of $V0_D$ neurons and lead to an increase in the population size of Evx1+ $V0_V$ neurons. As a secondary effect, the number of V2a neurons, which innervate the $V0_V$ neurons, declines at later developmental stages when neurons start to connect to each other, potentially compensating the increased number of $V0_V$ neurons. Only the subsets of interneuron populations are shown that are affected by the tissue-specific $Wt1$ knockout. Red indicates decrease in population size and green indicates increase in population size.

animals (below 1% for both). Thus, the increase in the amount of Evx1+ $V0_V$ neurons observed in mice lacking $Wt1$ does not seem to result from a direct transition of future Wt1+ dI6 neurons into Evx1+ $V0_V$ neurons.

## Discussion

In this study, we have examined Wt1, which marks a subset of dI6 neurons. We found that Wt1 is required for proper differentiation of spinal cord neurons during embryogenesis and that deletion of $Wt1$ results in locomotor aberrancies in neonates and adult mice.

Adult conditional $Wt1$ knockout animals ($Nes$-$Cre;Wt1^{fl/fl}$) show an increased stride length and a decreased stride frequency, resulting in slower absolute walking speed. This supports a possible role of the Wt1+ dI6 neurons in both timing and limitation of the stride cycle. Although the CPG network is capable of producing accurate timing and phasing, proprioceptive and supraspinal input is needed to regulate the CPG activity (Pearson, 2004; Kiehn, 2016; Danner et al, 2017). The integration of this sensory information would require an integrative position in the locomotor CPGs, which is compatible with the observed multisynaptic input to Wt1+ dI6 neurons. However, it still has to be determined whether these various inputs come from proprioceptive sensors, supraspinal regions, or other spinal cord interneurons.

The timing of hindlimb footfalls relative to forelimb footfalls during walking differed between $Wt1^{fl/fl}$ and $Nes$-$Cre;Wt1^{fl/fl}$ mice, particularly at the diagonal fore- and hindlimbs, suggesting that Wt1 cells play a role in long-range coordination between various spinal cord segments. We could show that at least a fraction of Wt1+ dI6 neurons possesses commissural projections, which is in accordance to very recent published data (Haque et al, 2018). Our results further revealed that deletion of $Wt1$ leads to a decline in the number of commissural neurons. This suggests not only an involvement of Wt1 in establishing proper projections of the $Wt1+$ dI6 neurons, but might also explain the changes in the phase coupling between contralateral limbs that might not be related to the V0-based interlimb coordination between fore- and hindlimbs (Talpalar et al, 2013; Danner et al, 2017).

The locomotor alterations observed in adult mice were more subtle than the locomotion abnormalities seen in neonates, which could be due to various compensatory adaptations during postnatal maturation of neuronal circuits. Neonates lacking Wt1 in the spinal cord increased the number of uncoordinated steps, which was supported by a markedly slower and variable pattern of locomotor-like activity in isolated $Nes$-$Cre;Wt1^{fl/fl}$ spinal cords. They also exhibited a perturbed flexor–extensor and left–right alternation that might be a consequence of the loss of commissural and ipsilateral projections. The data from the fictive locomotion

showing that a locomotor rhythm is established when *Wt1* is deleted suggested that Wt1+ dI6 neurons are unlikely to participate directly in the kernel of neurons that generate the locomotor rhythm (Dougherty et al, 2013). But because an increase in the variability of bursts during fictive locomotion was observed, we hypothesize that the Wt1+ dI6 neurons are involved in the modulation of this rhythm.

Lack of *Wt1* in the spinal cord caused alterations in the differentiation of dI6, V0, and V2a spinal cord neurons during embryogenesis (Fig 6C). The inverse alterations in the dI6 and V0 populations suggest a fate change from dI6 to V0-like neurons when *Wt1* is inactivated. The putative transition from dI6 to V0-like neurons occurs at the time point when *Wt1* expression would normally start. This instantaneous effect might be due to the derivation of both interneuron populations from neighboring progenitor domains sharing common transcription factors such as Dbx1 (Alaynick et al, 2011). Thus, loss of *Wt1* might lead to a switch in developmental programs that are normally repressed; whether this repression occurs cell-autonomously or non–cell-autonomously still has to be determined. In any case, when future *Wt1+* cells are ablated, an increase in V0-like neurons is no longer observed, suggesting that the fate switch requires the cells about to express *Wt1*.

The fate change of prospective dI6 to V0-like neurons is complex because dI6 neurons can be subdivided into at least three subsets based on the expression of the transcription factor–encoding genes *Wt1* and *Dmrt3* (Fig 6C). Loss of *Wt1* affects not only the small number of dI6 neurons that possess Wt1 and Dmrt3 but also the number of neurons that only express *Dmrt3*. This population is significantly decreased. The presence of Wt1+ dI6 neurons, therefore, is essential to maintain the character of a subset of Dmrt3+ dI6 neurons. If *Wt1* is inactivated, in addition to the cells that are programmed to express *Wt1*, possibly also Dmrt3+ dI6 neurons may differentiate into V0-like neurons.

Two main subpopulations exist within the V0 population (Alaynick et al, 2011): the Evx1+, more ventrally derived $V0_V$, and the Evx1 negative, more dorsally derived $V0_D$ population, for which no distinct marker has yet been described. The knockout of *Dbx1* results in trans-differentiation of the whole V0 population, whereby Evx1+ $V0_V$ neurons acquire a more ventral fate and become V1 neurons, whereas Evx1-negative $V0_D$ neurons acquire characteristics of dI6 neurons (Lanuza et al, 2004). This suggests that the $V0_D$, rather than the $V0_V$, neurons closely resemble the dI6 neurons and poses the question whether the fate change from Wt1+ dI6 neurons to Evx1+ $V0_V$-like neurons represents a direct or an indirect transition. The investigations using the *Nes-Cre;Wt1^{fl/GFP}* mice suggest that the *Wt1*-deficient dI6 cells do not change their fate directly into Evx1+ $V0_V$-like neurons, suggesting an indirect transition. This points to the possibility that the fate change might be achieved by a transition of Wt1+ dI6 neurons into the more closely related Evx1-negative $V0_D$-like neurons, which leads to a putative increase in the $V0_D$ population (Fig 6C). The Evx1+ $V0_V$ population might, in turn, increase its number to compensate for a higher proportion of $V0_D$-like neurons.

In addition to the changes in the dI6 and V0 populations that occur upon *Wt1* deletion in the spinal cord, Chx10+ V2a neurons show a slight but significant decrease in their cell number at E16.5 (Fig 6C). This might represent a secondary effect of the alterations in

the dI6 and V0 populations, which occur already at E12.5. It was reported that V2a neurons directly innervate $V0_V$ neurons (Crone et al, 2008). This secondary decrease in the number of these V2a neurons might thus be due to a potential adaptation to the increased number of $V0_V$ neurons.

The trans-differentiation of spinal cord neurons observed upon *Wt1* knockout might also have an effect on the locomotor phenotype detected in adults and neonates. Excitatory $V0_V$ and inhibitory $V0_D$ commissural neurons and the ipsilateral excitatory V2a interneurons built up a dual-inhibitory commissural system that is involved in left–right alternation of locomotion (Crone et al, 2008; Talpalar et al, 2013). This dual-inhibitory system works in a speed-dependent manner and allows switching between different gaits. At low speed, $V0_D$ neurons are active and cause walking and at higher speed, V2a and V0v neurons become active and cause trotting (Talpalar et al, 2013; Bellardita & Kiehn, 2015; Kiehn, 2016). In *Wt1* knockout animals, the trans-differentiation of neurons probably interferes with this system. This can be seen by the left–right perturbation observed in neonates, suggesting that the increase in V0 neurons is not sufficient to compensate for the loss of V2a neurons and commissural projections upon *Wt1* deletion. On the other hand, adult *Nes-Cre;Wt1^{fl/fl}* animals refused to run even when they were forced to move faster by increasing the speed of the treadmill. This suggests that they might have difficulties to run at high speeds and to switch from walk to other running gaits (Bellardita & Kiehn, 2015). However, their ability to change between different gaits still has to be investigated to elucidate the mechanism by which the increase in V0 neurons and the decrease in V2a neurons act in particular on the CPG output.

The approach to ablate a gene to investigate the function of a particular cell population in the spinal network comes with limitations. As already observed for the V0 neurons, inactivating a gene that is crucial for the differentiation of spinal cord neurons might lead to trans-differentiation. This is often accompanied by gain and loss of function of several neuron populations, which can make it challenging to assign distinct functions to a particular neuronal cell type (Lanuza et al, 2004; Talpalar et al, 2013). However, we have chosen the knockout of *Wt1* to investigate its role in the differentiation of spinal cord neurons and its influence on locomotion. If the scope was to determine the position of Wt1+ neurons in the locomotor CPG, silencing of the respective neurons would have been a more direct way as very recently performed by Haque et al (2018).. The authors report that acute silencing of these cells revealed their role for appropriate left–right alternation during locomotion (Haque et al, 2018). They also showed that Wt1+ dI6 neurons are inhibitory neurons with contralateral projecting axons terminating in close proximity to other commissural interneuron subtypes. Thus, our data and the results by Haque et al (2018) are complementary. We could not only confirm that Wt1+ dI6 neurons are commissural projecting but were also able to further show that Wt1 is crucial for the formation of these projections. Although the locomotor phenotype of the conditional *Wt1* knockout is more diverse than the altered left–right alternation seen in neonates with functionally silenced Wt1+-positive cells, the analyses of the locomotor behavior of adult *Wt1* knockout mice revealed a further involvement of Wt1+ dI6 neurons in modulating the gait rhythm. This modulation may be achieved because of an integrative position of the Wt1+ dI6 neurons with multisynaptic input.

In sum, the results obtained in this study shed light not only on the so far undescribed necessity for Wt1 in the development of spinal cord neurons but also on their functional implementation in circuits responsible for locomotion.

# Materials and Methods

### Mouse husbandry

All mice were bred and maintained in the Animal Facility of the Leibniz Institute on Aging—Fritz Lipmann Institute, Jena, Germany, according to the rules of the German Animal Welfare Law. Sex- and age-matched mice were used. Animals were housed under specific pathogen-free conditions, maintained on a 12-h light/dark cycle, and fed with mouse chow and tap water ad libitum. Mice used for analysis of fictive locomotion and projection tracing were kept according to the local guidelines of Swedish law. $Wt1^{fl/fl}$ mice were maintained on a mixed C57B6/J × 129/Sv strain. $Wt1$-GFP mice (Hosen et al, 2007) were maintained on a C57B6/J strain. Conditional $Wt1$ knockout mice were generated by breeding $Wt1^{fl/fl}$ females (Gebeshuber et al, 2013) to $Nes$-$Cre;Wt1^{fl/fl}$ (Tronche et al, 1999) or $Lbx1$-$Cre;Wt1^{fl/fl}$ mice (Sieber et al, 2007). To generate mice with Wt1-ablated cells, $Wt1$-$GFP$-$DTA$ mice were bred with $Lbx1$-$Cre$ mice. Control mice were sex- and age-matched littermates (wild type or $Wt1^{fl/fl}$). For plug mating analysis, females of specific genotypes were housed with males of specific genotypes and were checked every morning for the presence of a plug. For embryo analysis, pregnant mice were euthanized by $CO_2$ inhalation at specific time points during embryo development and embryos were dissected. Typically, female mice between 2 and 6 mo were used.

### Generation of Wt1-GFP-DTA mice

The $Wt1$-$GFP$-$DTA$ mouse line bares an $IRES$-$lox$-$GFP$-$lox$-$DTA$ cassette that was inserted into intron 3 of the $Wt1$ locus. This cassette consists of a $GFP$-encoding sequence that ends in a translational $STOP$ codon and is flanked by $loxP$ sites. Downstream of $GFP$, the coding sequence for the DTA was incorporated. Before Cre induction, the internal ribosomal entry site (IRES) cassette ensures the generation of a functional GFP protein. After Cre-mediated excision of the floxed $GFP$ sequence, the $DTA$ is expressed from the endogenous $Wt1$ promotor.

The $Wt1$-$GFP$-$DTA$ model was generated by homologous recombination in embryonic stem (ES) cells. After ES cell screening using PCR and Southern blot analyses, recombined ES cell clones were injected into C57BL/6J blastocysts. The injected blastocysts were reimplanted into OF1 pseudo-pregnant females and allowed to develop to term. The generation of F1 animals was performed by breeding of chimeras with wild-type C57BL/6 mice to generate heterozygous mice carrying the $Wt1$ knockin allele.

### Immunohistochemistry

Embryonic and postnatal spinal cords were dissected. They were either frozen unfixed after 15-min dehydration with 20% sucrose (in 50% TissueTec/PBS) (postfix) or fixed for 75 min in 4% para-formaldehyde in PBS (prefix). Prefixed tissue was cryoprotected in 10%, 20%, and 30% sucrose (in PBS) before freezing in cryoembedding medium (Neg-50; Thermo Fisher Scientific). Post- and prefixed samples were sectioned (12 $\mu$m). Postfixed samples were fixed for 10 min after sectioning and washed with 2% Tween in PBS (PBS-T). For prefixed samples, antigen retrieval was performed by incubation in sub-boiling 10 mM sodium citrate buffer (pH 6.0) for 30 min. After blocking with 10% goat serum and 2% BSA in PBS-T (postfix or prefix), the sections were incubated with primary antibodies (in blocking solution) using the following dilutions: gBhlhb5 1:50 (Santa Cruz Biotechnology, Inc.), BrdU 1:100 (abcam), shChx10 1:100 (abcam), gpDmrt3 1:5,000 (custom made [Andersson et al, 2012]), mEvx1 1:100 (1:3,000 prefix) (Developmental Studies Hybridoma Bank, University of Iowa), chGFP 1:1,000 prefix (abcam), mGFP 1:100 (Santa Cruz Biotechnology, Inc.), rFoxP2 1:800 (abcam), mIslet1/2 1:50 (Developmental Studies Hybridoma Bank, University of Iowa), gpLbx1 1:20,000 (gift from C. Birchmeier, MDC), Lim1/2 1:50 (Developmental Studies Hybridoma Bank, University of Iowa), NeuN 1:500 (Merck), rbPax2 1:50 (Thermo Fisher Scientific), rbLmx1b 1:100 (gift from R. Witzgall, University of Regensburg), and rbWt1 1:100 (Santa Cruz Biotechnology, Inc.). Secondary antibodies were applied according to species specificity of primary antibodies. Hoechst was used to stain nuclei. Quantitative analysis of the antibody staining was statistically analyzed using $t$ test.

### BrdU injection

To label proliferating cells in the embryonic spinal cord, pregnant mice at E9.5, E10.5, and E11.5 were injected intraperitoneally with 100 $\mu$g/g of BrdU dissolved in 0.9% sodium chloride solution. Embryos were harvested at E12.5 to isolate spinal cords and stain for BrdU and Wt1. Spinal cords were frozen unfixed after 15-min dehydration with 20% sucrose (in 50% TissueTec/PBS) and sectioned (12 $\mu$m). After any of the following treatments, the sections were washed with PBS. Antigen retrieval was performed by incubation in 98°C sub-boiling 10 mM sodium citrate buffer (pH 6.0) for 30 min. After treatment with 2N HCl at 37°C for 30 min, the sections were incubated with primary antibodies using the dilutions mentioned above (see the Immunohistochemistry section). Secondary antibodies were applied according to the species specificity of primary antibodies.

### RNA isolation and qRT–PCR analysis

Total RNA was isolated from E12.5 embryonic spinal cords using Trizol (Invitrogen) according to the manufacturer's protocol. Subsequently, 0.5 $\mu$g of RNA was reverse transcribed with iScript cDNA synthesis kit (Bio-Rad) and used for quantitative real-time PCR (qRT–PCR). The primer sequences used for RT–PCR analyses are as follows: TGT TAC CAA CTG GGA CGA CA ($Act\_for$); GGG GTG TGG AAG GTC TCA AA ($Act\_rev$); AGT TCC CCA ACC ATT CCT TC ($Wt1\_qRT\_for$); TTC AAG CTG GGA GGT CAT TT ($Wt1\_qRT\_rev$). Real-time PCR was carried out in triplicates for each sample using SyberGreenER (Thermo Fisher Scientific) and Bio-Rad iCycler (Bio-Rad). PCR efficiencies of primer pairs were calculated by the linear regression method. Ct values were normalized to the mean of the reference gene $Actin$.

Relative expression was determined by comparing normalized Ct values of *Wt1* conditional knockout and control samples (Pfaffl et al, 2002). Significance was determined by using pairwise reallocation randomisation test.

## Analysis of locomotor behavior

To characterize gait parameters, 10 animals per sex and genotype were used. Body masses of the mice varied considerably within the groups and among the groups with significant differences between the male *Wt1*$^{fl/fl}$ and *Nes-Cre;Wt1*$^{fl/fl}$ mice (*Wt1*$^{fl/fl}$: 28 g ± 3 g versus *Nes-Cre;Wt1*$^{fl/fl}$: 23 g ± 3 g; $F_s$ = 31.98; $t_s$ = 3.28, *P* > 0.001) and moderate differences between the female *Wt1*$^{fl/fl}$ and *Nes-Cre;Wt1*$^{fl/fl}$ mice (*Wt1*$^{fl/fl}$: 25 g ± 5 g versus *Nes-Cre;Wt1*$^{fl/fl}$: 22 g ± 4 g; $F_s$ = 3.80; $t_s$ = 1.62, not significant). We recorded the voluntary walking performance of this larger cohort using high-resolution X-ray fluoroscopy (biplanar C-arm fluoroscope Neurostar; Siemens AG). Strides defined as running gait according to the hindlimb duty factor (Hildebrand, 1985; Herbin et al, 2004) occasionally occur in some male *Wt1*$^{fl/fl}$ mice and were excluded from the analysis. Because of body size variation within and among groups, we adjusted treadmill speed dynamically to the individual preferences and abilities of the mice. This method of motion analysis has been described in detail in several recent publications (e.g., Böttger et al, 2011; Andrada et al, 2015; and Niederschuh et al, 2015) and will be only briefly summarized here. The X-ray system operates with high-speed cameras and a maximum spatial resolution of 1,536 dpi × 1,024 dpi. A frame frequency of 500 Hz was used. A normal-light camera operating at the same frequency and synchronized to the X-ray fluoroscope was used to document the entire trial from the lateral perspective. Footfall sequences and spatiotemporal gait parameters were quantified by manual tracking of the paw toe tips and two landmarks on the trunk (occipital condyles, iliosacral joint) using SimiMotion 3D. Speed, stride length, stride frequency, the durations of stance and swing phases, and the distances that trunk or limb covered during these phases were computed from the landmark coordinates collected at touchdown and liftoff of each limb. The phase relationships between the strides of left and right limbs as well as fore- and hindlimbs were determined from footfall sequences as expression of temporal interlimb coordination (Fig S1F). As the animals frequently accelerated or decelerated relative to the treadmill speed, the actual animal speed was obtained by offsetting trunk movement against foot movement during the stance phase of the limb. The resulting distance was divided by the duration of the stance phase. Animal speed and all temporal and spatial gait parameters were then scaled to body size following the formulas published by Hof (1996): nondimensional speed = v/$gl_0$, where v is raw speed, *g* is gravitational acceleration, and $l_0$ is the cube root of body mass as characteristic linear dimension, which scales isometrically to body mass; nondimensional frequency = f/$gl_0$, where f is raw frequency; and nondimensional stride length = l/$l_0$, where l is raw stride length. The scaled spatiotemporal gait parameters change as a function of nondimensional speed. Therefore, linear regression analyses were computed for each parameter in the male and the female *Wt1*$^{fl/fl}$ group. The power formulas obtained from regression computation (Y = a + bX) were then used to calculate the expected value for a given nondimensional speed for each gait

parameter (baseline) in each animal of all four groups. The coefficient of determination $r^2$ was computed. The deviations of the measured values of Y from the expected values, the residuals, were determined and are given in percent of deviation. Using these residuals, one-way ANOVA was computed to establish the significance of the differences between the means of *Wt1*$^{fl/fl}$ and *Nes-Cre;Wt1*$^{fl/fl}$ in males and females. Group means were calculated from the means of 10 animals. Sample size per mouse and limb ranged between 5 and 41 stride cycles, with an average sample size of 22 ± 9.

## Fictive locomotion

Animals (P0–P3) were euthanized and the spinal cords eviscerated in ice-cold cutting solution containing (in mM) 130 K-gluconate, 15 KCl, 0.05 EGTA, 20 Hepes, and 25 glucose (pH adjusted to 7.4 by 1M KOH) and then equilibrated in artificial cerebrospinal fluid (Perry et al, 2015) for at least 30 min before the beginning of experimental procedures. Suction electrodes were attached to left and right lumbar (L) ventral roots 2 and 5 (L2 and L5). A combination of NMDA (5 $\mu$M) + 5-HT (10 $\mu$M) + dopamine (50 $\mu$M) were added to the perfusing artificial cerebrospinal fluid to induce stable locomotor-like output. All chemicals were obtained from Sigma-Aldrich. Recorded signals containing compound action potentials were amplified 10,000 times and band-pass filtered (100–10 kHz) before being digitized (Digidata 1322A; Axon Instruments Inc.) and recorded using Axoscope 10.2 (Axon Instruments Inc.) for later off-line analysis. The data were rectified and low-pass filtered using a third-order Butterworth filter with a 5-Hz cutoff frequency before further analysis. Coherence plots between L2 and L2/L5 traces were analyzed using a mortlet wavelet transform in SpinalCore (Version 1.1). Preferential phase alignment across channels are shown in the circular plots and burst parameters were analyzed for at least 20 sequential bursts, as previously described (Kiehn & Kjaerulff, 1996) using an in-house designed program in Matlab (R2014b; Math-Works). Ventral root recording preferential phase alignment was assessed by means of circular statistics from five control cords and seven *Nes-Cre;Wt1*$^{fl/fl}$ cords (Rayleigh test and Watson's $U^2$ test) for 20 consecutive cycles as described (Kiehn & Kjaerulff, 1996). Burst parameters, including frequency, are presented as the mean ± SD. Burst parameters from five control cords and seven *Nes-Cre;Wt1*$^{fl/fl}$ cords were compared using the Mann–Whitney *U* test.

## Tracing of commissural neurons

To examine whether the loss of *Wt1* affects spinal cord populations, tracing experiments were conducted as previously described (Rabe et al, 2009; Andersson et al, 2012). *Nes-Cre;Wt1*$^{fl/fl}$ and *Nes-Cre;Wt1*$^{+/+}$ littermate control mice P0–P5 were prepared as described above (fictive locomotion). Two horizontal cuts (intersegmental tracing targeting commissural ascending/descending/bifurcating neurons) were made in the ventral spinal cord at lumbar (L) level 1 and between L4 and L5. Fluorescent dextran amine (FDA, 3,000 MW; Invitrogen) was applied at L1 and rhodamine dextran amine (RDA, 3,000 MW; Invitrogen) was applied between the L4/L5 ventral roots. Spinal cords were incubated overnight at room temperature, subsequently fixed in 4% formaldehyde, and stored in the dark at 4°C until transverse sectioning (60 $\mu$m) on a vibratome (Leica).

Fluorescent images were acquired on a fluorescence microscope (Olympus BX61W1). For quantitative analyses of traced cords, consecutive images were taken between the two tracer application sites using Volocity software (Improvision). Captured images were auto-leveled using Adobe Photoshop software. Only cords with an intact midline, as assessed during imaging, were used for analysis.

Traced neurons in $Wt1^{fl/fl}$ control and $Nes\text{-}Cre;Wt1^{fl/fl}$ cords were examined for significance using the Kruskal–Wallis analysis of variance test followed by a Dunns post-test comparing all groups. Tracing data are presented as the mean ± SEM using 3,975 total cells, 215 sections, and nine spinal cords ($Wt1^{fl/fl}$ control); 3,421 total cells, 228 sections, and seven spinal cords ($Nes\text{-}Cre;Wt1^{fl/fl}$).

### TUNEL assay

To detect apoptosis in situ, the TUNEL assay was performed before antibody binding. Slides were incubated with TUNEL reaction solution (1× reaction buffer TdT and 15 U TdT in ddH$_2$O from Thermo Fisher Scientific; 1 mM dUTP-biotin from Roche) at 37°C for 1 h and washed with PBS afterwards.

### Imaging and picture processing

Fluorescent images were viewed in a Zeiss Axio Imager and a Zeiss Observer Z1 equipped with an ApoTome slider for optical sectioning (Zeiss). Images were analyzed using the ZEISS ZEN2 image analysis software. For quantitative analyses of traced spinal cords, the application sites were identified and consecutive photographs were taken between the two application sites using the OptiGrid Grid Scan Confocal Unit (Qioptiq) and Volocity software (Improvision). Confocal images were captured on a ZEISS LSM 710 ConfoCor 3 confocal microscope and analyzed using the ZEISS ZEN2 image analysis software. Captured images were adjusted for brightness and contrast using ZEN2 image analysis software and Adobe Photoshop software.

### Statistical analyses

Data are expressed as mean ± SD or as mean ± SEM. Groups were compared using one-way ANOVA or two-tailed two-sample $t$ test depending on the number of groups and sample size. If normal distribution of a sample was not confirmed, sample means are compared by using nonparametric Mann–Whitney $U$ test. All statistical analyses were performed using GraphPad Prism Software (GraphPad Software Inc.), IBM SPSS Statistics 24 (IBM Corporation), Microsoft Excel (Microsoft Corporation), or Matlab (R2014b; MathWorks). Normal distribution was assessed using the D'Agostino-Pearson normality test or Kolmogorov–Smirnov test. Significance was determined as *$P < 0.05$, **$P < 0.01$, and ***$P < 0.001$.

### Treadmill gait analyses

This approach involved two groups of mixed sexes of $Wt1^{fl/fl}$ and $Nes\text{-}Cre;Wt1^{fl/fl}$ mice ($Wt1^{fl/fl}$: 10 males and 10 females; $Nes\text{-}Cre;Wt1^{fl/fl}$: 4 males and 8 females; age 24 wk). Locomotor performance was investigated at the German Mouse Clinic—Helmholtz Center, Munich, Germany (www.mouseclinic.de). Treadmill gait analysis was performed with the DigiGait Imaging System (Mouse Specifics, Inc.), which performs ventral plane videography to obtain digital footprints of a mouse walking on a transparent treadmill at different fixed speeds and subsequent analyses of gait patterns using DigiGait software. The DigiGait software determines treadmill contacts of individual paws that were used to quantify spatial (stride length) and temporal indices of gait parameters (stride, stance, and swing time) in walking or running animals. Paw placement of each limb is monitored throughout the gait cycle at up to 150 frames per second with a spatial resolution of more than 5,000 pixels per cm$^2$. For statistical analysis, at least 10 strides for each limb are included in the data set. The mean values for pairs of fore- and hindlimbs were used. Each speed was analyzed separately. Linear regression models using R (version 3.2.3) were used to determine the statistical significance between the groups (RCoreTeam, 2015). Because of strong influence of body weight and body length on the gait performance, those factors are also included into the model to dissect their combined effects on the data.

### Air-stepping analysis

P1 mice were used for air-stepping behavioral test according to Andersson et al (2012) (n = 9 $Wt1^{fl/fl}$ control, n = 7 $Nes\text{-}Cre;Wt1^{fl/fl}$) to investigate limb movements in neonates. The hindlimb steps were manually recorded for each animal over 20 s. The number of alternating, synchronous, and uncoordinated hindlimb steps was determined. Each parameter was then statistically tested using the $t$ test. The experimenter was blind to genotype while performing and analyzing experiments.

## Supplementary Information

## Acknowledgements

We thank D Kruspe and R Peterson for technical assistance; C. Birchmeier (Max Delbrück Center for Molecular Medicine, Berlin, Germany) for providing the $Lbx1\text{-}Cre$ mouse line; and C Hübner, H Heuer, and G Zimmer for critical discussion. This project was supported by grants from the German Federal Ministry of Education and Research (Infrafrontier grant 01KX1012) to L Becker and the Swedish Medical Research Council, Hållsten, Ländells, Swedish Brain Foundations to K Kullander. D Schnerwitzki received a scholarship from the Leibniz Graduate School on Ageing and Age-Related Diseases. FV Caixeta was funded by a scholarship from CNPqi–Brazil. The Fritz Lipmann Institute is a member of the Leibniz Association and is financially supported by the Federal Government of Germany and the State of Thuringia.

### Author Contributions

D Schnerwitzki: conceptualization, formal analysis, investigation, visualization, and writing—original draft, review, and editing.
S Perry: investigation, visualization, and writing—original draft, review, and editing.
A Ivanova: conceptualization and investigation.

F Caixeta: formal analysis, methodology, and writing—review and editing.

P Cramer: investigation and visualization.

S Guenther: investigation.

K Weber: investigation.

A Tafreshiha: investigation.

L Becker: formal analysis, supervision, and visualization.

I Vargas Panesso: investigation.

T Klopstock: supervision and project administration.

M Hrabě de Angelis: supervision and project administration.

M Schmidt: conceptualization, resources, formal analysis, supervision, investigation, visualization, methodology, and writing—original draft, review, and editing.

K Kullander: conceptualization, supervision, project administration, and writing—review and editing.

C Englert: conceptualization, supervision, funding acquisition, project administration, and writing—original draft, review, and editing.

## Conflict of Interest Statement

The authors declare that they have no conflict of interest.

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
