## [Reviewer comments · Life Science Alliance]

Neuron-specific inactivation of *Wt1* alters locomotion in mice and changes interneuron composition in the spinal cord

Danny Schnerwitzki, Sharn Perry, Anna Ivanova, Fabio Viegas Caixeta, Paul Cramer, Sven Günther, Kathrin Weber, Atieh Tafreshiha, Lore Becker, Ingrid L. Vargas Panesso, Thomas Klopstock, Martin Hrabe de Angelis, Manuela Schmidt, Klas Kullander, Christoph Englert
DOI: 10.26508/lsa.201800106

Review timeline:

Submission Date:	12 June 2018
Editorial Decision:	20 June 2018
Revision Received:	20 July 2018
Editorial Decision:	25 July 2018
Revision Received:	09 August 2018
Accepted:	10 August 2018

Report:

(Note: Letters and reports are not edited. The original formatting of letters and referee reports may not be reflected in this compilation.)

1st Editorial Decision

20 June 2018

Your manuscript was assessed by a really good expert from the field who provided constructive input. As you will see, this reviewer thinks that acute ablation of WT1 neurons as performed in the related work recently published would have been a better approach. However, there is clearly merit in your study, and the reviewer offers input on how you could address this concern and other issues in a minor revision, with text changes acknowledging caveats and by providing information you should already have at hand.

We have discussed your work in light of these comments and decided that it is warranted to publish your work pending such minor revision. We would therefore like to ask you to address the individual concerns the reviewer notes and to provide a revised version. Please get in touch in case you would like to discuss individual concerns further. As discussed with you by telephone before, the related work should also get cited and discussed when revising the manuscript.

REFeree REPORTS

Reviewer #1 (Comments to the Authors (Required)):

This manuscript by Schnerwitzki et al. aims to address the role of *Wt1* expressing neurons' involvement in generating locomotion and *Wt1*'s role for development of spinal interneurons. The authors determine that the *Wt1* spinal population belongs to the *di6* neurons that have previously been characterized partly by looking at *Dmrt3* positive neurons (by the lab of Kullander). Now, using knock out of the *Wt1* gene, the authors show that locomotion recorded in adult *Wt1* knockout mice is slightly affected and that drug-induced locomotor-like activity in the isolated spinal cord becomes irregular. They also show that *Wt1*⁺ neurons in the cord project commissurally and received inputs from excitatory and inhibitory neurons of unknown origin. They finally show, that the *Wt1* gene is fate determining and that absence of the gene leads to trans-differentiation with an increased number of *V0V* neurons and decreased number of *V2a* neurons with no changes in other spinal interneuron populations.

While, I am generally sympathetic to the study and agree that addressing the functional role of the *Wt1*⁺ population in the context of locomotion is interesting, I feel that the locomotor analysis is

hampered by methodological constraints that makes the interpretation of the findings difficult. Moreover, these findings are not placed in a proper context of the current knowledge about the vertebrate locomotor networks and their function. These issues are outlined below together with other comments that should improve the presentation.

- 1) Knocking out the gene in the spinal population might be of interest e.g. when comparing the role of the *Dmrt3* gene for the function of the locomotor network in the mouse phenotype was compared to horse gene-mutant phenotype (Andersson et al. 2012). However, if the scope is to determine the functional role of a cell population in the spinal network it is not the method of choice. The reason is that cells will lose their gene of interest but might not disappear from the network (which is difficult to determine and was only indirectly in the present study) and the lack of the gene might lead to trans-differentiation. The latter effect was clearly demonstrated in the present study, with an increase in V0V neurons a decrease in *Dmrt3* and V2a neurons. These mixtures of (undetermined) loss of the target neuron population and gain and loss of function of other neuronal populations makes it difficult to assign to the target population. In the particular case the decrease in *Chx10* neurons by *Wt1* gene knockout might lead to change in left-right alternation and burst amplitude (Crone et al. 2008, 2009; Dougherty et al. 2013) effect that are not directly related to the loss of *Wt1+* cells. The decrease of *Dmrt3+* neurons will also affect the phenotype as previously reported. Therefore, the loss of a neuronal type is best studied using acute or chronic silencing of the *Wt1+* population using a *Wt1-cre* (available from Jax) mice in combination with DAT, Tetanus light chain, Halorhodopsin or inhibitory DREADDs as have been done in many recent locomotor studies for the V0-V3 ladder (Gosgnach et al. 2006; Crone et al. 2008, 2009; Talpalar et al. 2013; Dougherty et al. 2013; Zhang et al. 2008, see Goulding 2009 and Kiehn 2016; Arber 2008). These experiments could have been done *in vitro* (also in late prenatal animals) or *in vivo* if combined with intersectional deletion in the spinal cord as performed by Goulding et al. The need for this type of experiments is enhanced by a newly published study in *J. Neuroscience* (that the authors do not cite) - *WT1*-expressing interneurons regulate left-right alternation during mammalian locomotor activity - by Farhia Haque et al. 2018, which uses acute silencing *in vitro* where they only perturb the activity in the *Wt1+* population with clear effect on the left-right coordination. The effects on locomotor parameters are much more mixed in the present study (with effect on both left-right and flexor extensor coordination, burst duration and frequency). Given that the results of the two papers will be compared it is important to acknowledge the limitations of the gene-ablation approach and the confounding factors (see also Lanuza et al. *Neuron* 2004 (gene ablation of V0 neurons) as compared to Talpalar et al. *Nature* 2013 (ablation of V0 neurons)). If the authors do not want to perform target deletion studies which could be extended to *in vivo* studies and thereby extend on the Haque study they need to discuss thoroughly the confounding factors in direct relationship to the Haque paper. From the data presented in Fig. 2 and 4 it is very difficult to extract a specific role for *Wt1* neurons in the network. My concern this might not be real. These cells might indeed have a specific role in the network that will be revealed by target deletion as done in the Haque paper.
 - 2) When knocking out the gene in the entire animals it is not only the spinal cord that is affected. The authors need to acknowledge if the *Wt1* gene is expressed in other areas of the nervous system.
 - 3) The *in vivo* analysis of locomotion is suboptimal. Since mice can change gaits (Bellardita and Kiehn, 2015) it will be informative to know if the gaits were the same throughout the different speeds of locomotion both on the treadmill and when the animals were moving spontaneously. Is it walk or slow trot? If both are present I suggest that the different gaits are divided before looking for coordination changes.
 - 4) For the *in vitro* analysis: please test statistically if there is a difference between the knockout and WT mice in left-right and flexor extensor coordination (Wilson's test).
 - 5) Page 9: while it is of interest to know that the cells have different inputs it should be acknowledged that it is unknown from where these inputs come. Some sort of quantification would also be advantageous. Fig. 4 E-H is nice and important because it indicates that commissural *WT1* neurons are indeed lost by the gene-deletion (especially with the increase in the V0V population). This finding should be enhanced in wording because it provides explanation to the left-right perturbation that is seen *in vitro* (together with the decreased V2a population).
 - 6) It is a major limitation to the interpretation to the functional role of the cells that the authors do not provide the transmitter-phenotype for the *Wt1+* cells.
 - 7) The introduction and discussion and the conceptual framework for the study are not placed in a proper context of the current knowledge about the vertebrate locomotor networks and their function. The references to the locomotor networks should be updated and supplemented.
- Introduction line 2: change references to Grillner 2003, Kiehn 2006, Goulding 2009; Brownstone

and Wilson 2008; Roberst et al. 2010 etc. same last line. Delete Shik and Orlovsky 1975. Page 4 line 4 add Jessell 2000. Last line first para: add Kiehn 2006, 2016, Goulding 2009; Brownstone and Wilson 2008; Grillner and Jessell 2009. Next paragraph V0-V3 neurons in locomotion: add references to Lanuza et al. 2004, Talpalar et al 2013, Belardita and Kiehn 2015 (V0), V1 Zhang et al. 2014, Britz et al. 2015 (V1), Crone et al. 2008, 2009, Dougherty and Kiehn 2010, Zhong et al. 2010 (V2a) and Zhnag et al. 2008 (V3). This section should also refer to the recent published paper on Wt1 and remove the sentence saying that WT1 has not been investigated.

Discussion and the conceptual framework: The authors talk about left-right, flexor and extensor and speed, but they are very vague when discussing what these parameters means. When discussing the results, I suggest that they are much more specific about these parameters and what we know about the networks involved in controlling them (see Kiehn 2016). What does it mean that the Wt1 cells have an 'involvement in timing and limitation of the stride cycle'? What is the mechanism for this given our current knowledge of the mammalian spinal locomotor network? The small change in phase-coupling between diagonal fore and hindlimbs is interesting and might be related to a specific loss of WT1 commissural neurons that is not related to the V0 secured left-right alternation and coordination between fore and hindlimbs (Talpalar et al. 2013 and Danner et al. 2017). A better discussion of what is known about the role of the broader V0 population (V0D and V0V) and the V2a neurons in securing left right alternation with reference to the work by Talpalar et al. Crone et al. Dougherty et al. Zhong et al. is needed. The last sentence in the discussion is to bold.

Minor changes

1. Title needs to be changes - the deletion is not specific to the spinal cord.
2. Page 7: Raw speeds?
3. Temporal coordination between stance and swing phase: that speed dependent and needs to be shown as such if any claims are made.
4. Page 13: motor cortex does not override spinal circuits-especially not during locomotion.
5. Figure 1E: quantification (number of cells) is needed.

1st Revision – authors' response

20 July 2018

We are grateful for the very detailed and constructive reviewer's comments. In the following we describe how we have addressed the individual points:

1) Knocking out the gene in the spinal population might be of interest e.g. when comparing the role of the Dmrt3 gene for the function of the locomotor network in the mouse phenotype was compared to horse gene-mutant phenotype (Andersson et al. 2012). However, if the scope is to determine the functional role of a cell population in the spinal network it is not the method of choice. The reason is that cells will loose their gene of interest but might not disappear from the network (which is difficult to determine and was only indirectly in the present study) and the lack of the gene might lead to trans-differentiation. The latter effect was clearly demonstrated in the present study, with an increase in V0V neurons a decrease in Dmrt3 and V2a neurons. These mixtures of (undetermined) loss of the target neuron population and gain and loss of function of other neuronal populations makes it difficult to assign to the target population. In the particular case the decrease in Chx10 neurons by Wt1 gene knockout might lead to change in left-right alternation and burst amplitude (Crone et al. 2008, 2009; Dougherty et al 2013) effect that are not directly related to the loss of Wt1+ cells. The decrease of Dmrt3+ neurons will also affect the phenotype as previously reported. Therefore, the loss of a neuronal type is best studied using acute or chronic silencing of the Wt1+ population using a Wt1-cre (available form Jax) mice in combination with DAT, Tetanus light chain, Halorhodopsin or inhibitory DREEDs as have been done in many recent locomotor studies for the V0-V3 ladder (Gosnack et al. 2006; Crone et al. 2008, 2009; Talpalar et al. 2013; Dougherty et al. 2013; Zhang et al. 2008, see Goulding 2009 and Kiehn 2016; Arber 2008). These experiments could have been done in vitro (also in late prenatal animals) or in vivo if combined with intersectional deletion in the spinal cord as performed by Goulding et al. The need for this type of experiments is enhanced by a newly published study in J. Neuroscience (that the authors do not cite) - WT1-expressing interneurons regulate left-right alternation during mammalian locomotor activity - by Farhia Haque et al. 2018, which uses acute silencing in vitro where they only perturb the activity in the Wt1+ population with clear effect on the left right coordination. The effects on locomotor parameters are much more mixed in the present study (with effect on both left-right and flexor extensor coordination, burst duration and frequency). Given that the results of the two papers

will be compared it is important to acknowledge the limitations of the gene-ablation approach and the confounding factors (see also Lanuza et al. Neuron 2004 (gene ablation of V0 neurons) as compared to Talpalar et al. Nature 2013 (ablation of V0 neurons)). If the authors do not want to perform target deletion studies which could be extended to in vivo studies and thereby extend on the Haque study they need to discuss thoroughly the confounding factors in direct relationship to the Haque paper. From the data presented in Fig. 2 and 4 it is very difficult to extract a specific role for Wt1 neurons in the network. My concern this might not be real. These cells might indeed have a specific role in the network that will be revealed by target deletion as done in the Harque paper.

We agree with the reviewer that the ablation of a gene to investigate the function of a particular cell population in the spinal cord network expressing the gene only allows limited insights into the functional role of the cell. Especially when the knockout leads to transdifferentiation that is accompanied by gain and loss of function of several neuron populations as seen for the *Wt1* knockout. This can make it difficult to assign the position of *Wt1*+ cells in the locomotor CPG. However, our scope was not to determine the position of *Wt1*+ neurons in the locomotor CPG. Our aim was to investigate the role of *Wt1* in the differentiation of spinal cord neurons and its influence on locomotion.

We admit in the discussion that silencing of the respective neurons is a more direct way to address the functional role of these cells in the CPG as very recently performed by Haque *et al.* We discuss the findings of Haque *et al.* and interpret our data in comparison to theirs.

2) When knocking out the gene in the entire animals it is not only the spinal cord that is affected. The authors need to acknowledge if the Wt1 gene is expressed in other areas of the nervous system.

Since a constitutive knockout of *Wt1* is embryonically lethal, we made use of a conditional *Nes-Cre; Wt1^{fl/fl}* mouse line to investigate the function of *Wt1* in the spinal cord. The *Nes-cre* is expressed in neuronal progenitor cells.

Armstrong et al. and Rackley et al. reported about expression of *Wt1* in two areas of the CNS namely the embryonic spinal cord and a particular region of the hindbrain below the 4th ventricle. The fact that *Wt1* is expressed in the hindbrain region has been added in the introduction.

As a note for the reviewer: we checked for deletion of *Wt1* in both regions of *Wt1* expression in the CNS and confirmed deletion of *Wt1* only in the spinal cord but not in the region of the hindbrain below the 4th ventricle.

3) The in vivo analysis of locomotion is suboptimal. Since mice can change gaits (Bellardita and Kiehn, 2015) it will be informative to know if the gaits were the same throughout the different speeds of locomotion both on the treadmill and when the animals were moving spontaneously. Is it walk or slow trot? If both are present I suggest that the different gaits are divided before looking for coordination changes.

We identified gaits according to the well-established gait definitions introduced by Milton Hildebrand in the 1960ies, which have been already used to study the gaits of mice by Herbin et al (2004). Accordingly, the Duty factor of the hindlimb or sometimes the average Duty factor of fore- and hindlimb, are used to distinguish walking and running within symmetrical gaits. In our study, we excluded all running cycles from the sample, which we occasionally recorded in male *Wt1*fl/fl. We therefore only analyzed walk.

4) For the in vitro analysis: please test statistically if there is a difference between the knockout and WT mice in left-right and flexor extensor coordination (Wilson's test).

We tested statistically if there is a difference between the knockout and control animals in left-right and flexor extensor coordination by using the Watson's U2 test. The respective result is added in the text.

5) Page 9: while it is of interest to know that the cells have different inputs it should be acknowledged that it is unknown from where these inputs come. Some sort of quantification would also be advantageous. Fig. 4 E-H is nice and important because it indicates that commissural WT1 neurons are indeed lost by the gene-deletion (especially with the increase in the VOV

population). *This finding should be enhanced in wording because it provides explanation to the left-right perturbation that is seen in vitro (together with the decreased V2a population).*

We have acknowledged in the discussion that it still has to be determined where the various synaptic inputs of the Wt1+ neurons come from.

A quantification of the synapses connecting to the Wt1+ neurons was not carried out due to technical reasons. However, we do not see this as a disadvantage because the qualitative analyses already showed that the Wt+ neurons receive various inputs. In our opinion, an additional quantification of the synapses would not lead to a better understanding about the primary source of innervation of the Wt1+ neurons.

The decreased number of commissural projections in conditional Wt1 knockout animals was emphasized and put into context with the alterations in the V0v and V2a population as well as the left-right perturbation.

6) It is a major limitation to the interpretation to the functional role of the cells that the authors do not provide the transmitter-phenotype for the Wt1+ cells.

We have not determined the transmitter phenotype of the Wt1+ cells. We agree that this is a limitation of our work. However, the transmitter type of Wt1+ cells has meanwhile been determined as GABAergic (70%) and glycinergic (30%) by Haque et al.

7) The introduction and discussion and the conceptual framework for the study are not placed in a proper context of the current knowledge about the vertebrate locomotor networks and their function. The references to the locomotor networks should be updated and supplemented.

Introduction line 2: change references to Grillner 2003, Kiehn 2006, Goulding 2009; Brownstone and Wilson 2008; Roberst et al. 2010 etc. same last line. Delete Shik and Orlovsky 1975. Page 4 line 4 add Jessell 2000. Last line first para: add Kiehn 2006, 2016, Goulding 2009; Brownstone and Wilson 2008; Grillner and Jessell 2009. Next paragraph V0-V3 neurons in locomotion: add references to Lanuza et al. 2004, Talpalar et al 2013, Belardita and Kiehn 2015 (V0), VI Zhang et al. 2014, Britz et al. 2015 (VI), Crone et al. 2008, 2009, Dougherty and Kiehn 2010, Zhong et al. 2010 (V2a) and Zhnag et al. 2008 (V3). This section should also refer to the recent published paper on Wt1 and remove the sentence saying that WT1 has not been investigated.

All the references to the locomotor CPG have been updated and supplemented according to the reviewer's suggestions.

Discussion and the conceptual framework: The authors talk about left-right, flexor and extensor and speed, but they are very vague when discussing what these parameters means. When discussing the results, I suggest that they are much more specific about these parameters and what we know about the networks involved in controlling them (see Kiehn 2016). What does it mean that the Wt1 cells have an 'involvement in timing and limitation of the stride cycle'? What is the mechanism for this given our current knowledge of the mammalian spinal locomotor network? The small change in phase-coupling between diagonal fore and hindlimbs is interesting and might be related to a specific loss of WT1 commissural neurons that is not related to the V0 secured left-right alternation and coordination between fore and hindlimbs (Talpalar et al. 2013 and Danner et al. 2017). A better discussion of what is known about the role of the broader V0 population (V0D and V0V) and the V2a neurons in securing left right alternation with reference to the work by Talpalar et al. Crone et al. Dougherty et al. Zhong et al. is needed. The last sentence in the discussion is to bold.

We appreciate the suggestions and literature hints that the reviewer made about the discussion and the conceptual framework. We discussed the points mentioned by re-writing the discussion and answering the reviewer's questions.

Minor changes

1. *Title needs to be changes - the deletion is not specific to the spinal cord.*

Title was changed to avoid misinterpretation that Wt1 deletion would be specific to the spinal cord.

2. *Page 7: Raw speeds?*

Raw speed is animal velocity in m/s while size-corrected speed is a dimensionless parameter. This definition was added in the text.

3. *Temporal coordination between stance and swing phase: that speed dependent and needs to be shown as such if any claims are made.*

For the speed dependence of duty factor, the F-value and the coefficient of determination have been provided.

4. *Page 13: motor cortex does not override spinal circuits-especially not during locomotion.*

The statement that the motor cortex would override spinal circuits was removed.

5. *Figure 1E: quantification (number of cells) is needed.*

Regarding quantification it was emphasized in the text that all the Wt1 cells in the spinal cord are positive for the respective dl6 markers: Pax2, Lim1/2, Lbx1 and Bhlhb5.

2nd Editorial Decision

25 July 2018

Thank you for submitting your revised manuscript entitled "Neuron-specific inactivation of Wt1 alters locomotion in mice and changes interneuron composition in the spinal cord". The original reviewer assessed your manuscript again and now supports publication in Life Science Alliance, pending final revision. We would thus like to invite you to address this reviewer's remaining points - appended below - by editing the manuscript slightly. Please also include the supplementary method section in the main manuscript, and add n and statistical test used / what kind of error bar is shown to the figure legends. Finally, it is unclear whether the wildtype staining in figure 4D is an enlarged inset or not (please clarify and add source image if it is).

REFEREE REPORTS

Reviewer #1 (Comments to the Authors (Required)):

The authors have done a good job in revising the paper and including most of my suggestions.

There is still a few issues left that need editorial attention:

- 1) Page 4 second section, line 4: please add Jessell 2000.
- 2) Page 4 section 1: coordinated pattern not coordinated rhythm
- 3) Section on in vivo locomotion and related discussion about speed. It is mentioned in the discussion that the Wt-Knockouts cannot perform running gaits (pages 14 and 17). While this may be true there is no hard data to back up this claim. You don't show that wt can do that and there is no strong test to exclude that knockouts cannot (see for example Belardita and Kiehn 2015, or

Caggiano et al. 2018). You might be right but as it is it you should be careful to state it. I suggest to tone down these statements unless you can provide strong experimental evidence to support the claim.

- 4) It is a bit hard for persons that are not used to look at the phase diagram data to capture the message. Suggest to indicate clearly in a diagram what phases means (footfall to footfall or end of stance phase to end of stance phase..)
- 5) In the Circular plots for in vitro - please indicate the significance level in the circles (not only the high).
- 6) Page 14, para 1 last sentence: why can they not come from interneurons?
- 7) Page 15 line 5: the in vitro prep does not produce steps.
- 8) Page 15 line 8: ...commissural and ipsilateral projection.
- 9) Page 15 first section. Add a reference to paper that discuss rhythm generation: for example Dougherty et al. 2013 and mention what you mean by rhythm generation.
- 10) Page 15, line 12: delete 'putative by
- 11) Page 16, second para, line 4: replace loss with trans-differentiation
- 12) Page 17, line7: it is the other way around. V2a and V0v are active at higher speeds producing trot while V0d are active at lower speeds producing walk Talpalar et al. 2013, Belardita and Kiehn 2015 and Kiehn 2016).

2nd Revision – authors' response

09 August 2018

We are again grateful for the constructive comments. In the following we describe how we have addressed the individual points:

- 1) *Page 4 second section, line 4: please add Jessell 2000.*

The reference was added.

- 2) *Page 4 section 1: coordinated pattern not coordinated rhythm*

The word rhythm was replaced by pattern leading to the phrase “which generate an organized motor pattern during repetitive locomotor tasks”.

- 3) *Section on in vivo locomotion and related discussion about speed. It is mentioned in the discussion that the Wt-Knockouts cannot perform running gaits (pages 14 and 17). While this may be true there is no hard data to back up this claim. You don't show that wt can do that and there is no strong test to exclude that knockouts cannot (see for example Belardita and Kiehn 2015, or Caggiano et al. 2018). You might be right but as it is it you should be careful to state it. I suggest to tone down these statements unless you can provide strong experimental evidence to support the claim.*

The statement was toned down and we now mention that the ability of knockouts to change between different gaits still has to be investigated.

- 4) *It is a bit hard for persons that are not used to look at the phase diagram data to capture the message. Suggest to indicate clearly in a diagram what phases means (footfall to footfall or end of stance phase to end of stance phase..)*

We have added a diagram (see supplemental Figure 1F) to explain the meaning of phases and of the terms related to it.

- 5) *In the Circular plots for in vitro - please indicate the significance level in the circles (not only the high).*

The significance level has been indicated in the circular plots.

- 6) *Page 14, para 1 last sentence: why can they not come from interneurons?*

The reviewer is completely right. It is also possible that dl6 interneurons receive input from other spinal cord interneurons.

7) Page 15 line 5: *the in vitro prep does not produce steps.*

We did not mean the *in vitro* preparation to produce steps but the air stepping experiment. This statement was added to the text.

8) Page 15 line 8: *...commissural and ipsilateral projection.* ,

We have included ipsilateral projections in the explanation.

9) Page 15 first section. *Add a reference to paper that discuss rhythm generation: for example Dougherty et al. 2013 and mention what you mean by rhythm generation.*

As suggested the respective reference was added and we now explain more specifically what we mean regarding participation of the Wt1+ neurons in the modulation of rhythm generation.

10) Page 15, line 12: *delete 'putative by'*

The phrase “putatively by connecting different contralateral rhythm generators” was deleted.

11) Page 16, second para, line 4: *replace loss with trans-differentiation*

The word “loss” was replaced with “trans-differentiation”.

12) Page 17, line 7: *it is the other way around. V2a and V0v are active at higher speeds producing trot while V0d are active at lower speeds producing walk Talpalar et al. 2013, Belardita and Kiehn 2015 and Kiehn 2016).*

We apologize for this *faux pas* and corrected the mistake.

3rd Editorial Decision

10 August 2018

Thank you for submitting your Research Article entitled "Neuron-specific inactivation of Wt1 alters locomotion in mice and changes interneuron composition in the spinal cord". It is a pleasure to let you know that your manuscript is now accepted for publication in Life Science Alliance. Congratulations on this interesting work.